# Genome-wide prediction of synthetic rescue mediators of resistance to targeted and immunotherapy

Avinash Das Sahu[1,2,3,*,†] (iD), Joo S Lee[3,4,†,‡] (iD), Zhiyong Wang[5], Gao Zhang[6,7], Ramiro Iglesias-Bartolome[8,‡] (iD), Tian Tian[9] (iD), Zhi Wei[9], Benchun Miao[2], Nishanth Ulhas Nair[3,4,‡], Olga Ponomarova[10], Adam A Friedman[2], Arnaud Amzallag[2], Tabea Moll[2], Gyulnara Kasumova[2], Patricia Greninger[2], Regina K Egan[2], Leah J Damon[2], Dennie T Frederick[2], Livnat Jerby-Arnon[11], Allon Wagner[12], Kuoyuan Cheng[3], Seung Gu Park[1], Welles Robinson[3], Kevin Gardner[4,‡], Genevieve Boland[2], Sridhar Hannenhalli[3], Meenhard Herlyn[6], Cyril Benes[2], Keith Flaherty[2], Ji Luo[8,‡], J Silvio Gutkind[5] & Eytan Ruppin[3,4,11,**,‡] (iD)

## Abstract

Most patients with advanced cancer eventually acquire resistance to targeted therapies, spurring extensive efforts to identify molecular events mediating therapy resistance. Many of these events involve *synthetic rescue (SR) interac*tions, where the reduction in cancer cell viability caused by targeted gene inactivation is rescued by an adaptive alteration of another gene (the *rescuer*). Here, we perform a genome-wide *in silico* prediction of SR rescuer genes by analyzing tumor transcriptomics and survival data of 10,000 TCGA cancer patients. Predicted SR interactions are validated in new experimental screens. We show that SR interactions can successfully predict cancer patients' response and emerging resistance. Inhibiting predicted rescuer genes sensitizes resistant cancer cells to therapies synergistically, providing initial leads for developing combinatorial approaches to overcome resistance proactively. Finally, we show that the SR analysis of melanoma patients successfully identifies known mediators of resistance to immunotherapy and predicts novel rescuers.

**Keywords** drug combination; drug resistance; immunotherapy; synergy
**Subject Categories** Cancer; Computational Biology; Genome-Scale & Integrative Biology

**Mol Syst Biol.** (2019) 15: e8323

## Introduction

Despite major advances in cancer therapies, many patients eventually succumb to emerging resistance. Recent experimental and clinical studies have successfully characterized tumor-specific molecular signatures of resistance to targeted therapies through DNA and RNA sequencing (Jones *et al*, 2012; MacArthur *et al*, 2014; Bertotti *et al*, 2015; Fong *et al*, 2015; Miyamoto *et al*, 2015; Rathert *et al*, 2015; Wilson *et al*, 2015; Raphael *et al*, 2017). However, these studies require an arduous collection and molecular profiling of paired pre- and post-treatment tumor biopsies (Beltran *et al*, 2016) and cannot be conducted for drugs at early stages of their development. Thus, the development of a computational approach that can expedite the identification of resistance determinants from existing large-scale cancer cohorts' data is warranted.

To this end, we have set out to predict *synthetic rescue* (SR) interactions (Motter *et al*, 2008; Fong et al, 2015; Miyamoto *et al*, 2015; Rathert *et al*, 2015; van Leeuwen *et al*, 2016), which are a

1 Department of Biostatistics and Computational Biology, Harvard School of Public Health, Boston, MA, USA
2 Department of Medicine and Harvard Medical School, Massachusetts General Hospital Cancer Center, Boston, MA, USA
3 University of Maryland Institute of Advanced Computer Science (UMIACS), University of Maryland, College Park, MD, USA
4 Cancer Data Science Lab, National Cancer Institute, National Institutes of Health, Bethesda, MD, USA
5 Department of Pharmacology & Moores Cancer Center, University of California, San Diego, La Jolla, CA, USA
6 Molecular and Cellular Oncogenesis Program and Melanoma Research Center, The Wistar Institute, Philadelphia, PA, USA
7 Department of Neurosurgery and The Preston Robert Tisch Brain Tumor Center, Duke University, Durham, NC, USA
8 National Cancer Institute, National Institutes of Health, Bethesda, MD, USA
9 New Jersey Institute of Technology, Newark, NJ, USA
10 University of Massachusetts Medical School, Worcester, MA, USA
11 Schools of Computer Science & Medicine, Tel-Aviv University, Tel-Aviv, Israel
12 Department of Electrical Engineering and Computer Science, the Center for Computational Biology, University of California, Berkeley, CA, USA
 *Corresponding author. Tel: +1 240 391 8125; E-mail: asahu@jimmy.harvard.edu
 **Corresponding author. Tel: +1 240 858 3169; E-mail: eyruppin@gmail.com
 †These authors contributed equally to this work
 ‡This article has been contributed to by US Government employees and their work is in the public domain in the US

generalization of suppressor interactions (Szappanos et al, 2011a). Suppressor interactions, recently identified in yeast genome-wide (van Leeuwen et al, 2016), denote a functional interaction where following the inactivation of specific genes, cells suppress additional genes to escape from harmful alterations (Bouwman et al, 2010; Xu et al, 2015; Forment et al, 2017). SR interactions denote a functional interaction where a fitness reducing alteration due to inactivation of one gene (termed the vulnerable gene) is compensated by altered activity (downregulation or upregulation) of another, rescuer gene (Papp et al, 2003; Kafri et al, 2005, 2009; Beltran et al, 2016) (Fig 1A). As rescue events are required to compensate for fitness reducing alterations occurring during the natural evolution of cancer (Szamecz et al, 2014), one may expect to detect the SR interactions forged in evolving tumors, even untreated ones (Landau et al, 2013; Taylor-Weiner et al, 2016; Carter et al, 2017). When a vulnerable gene is targeted by an anti-cancer drug (Hart et al, 2015), such SR interactions may manifest by changes in the activity of its interacting rescuer gene(s), thus mediating drug resistance. Both primary and adaptive resistance could be mediated by SR mechanisms.

We have recently developed a data mining approach, ISLE (Lee et al, 2018), that mines TCGA and published in vitro screens to identify clinically relevant synthetic lethal (SL) interactions. An SL gene pair when co-inactive exhibits negative selection as it decreases tumor fitness. ISLE harnesses this principle to identify gene pairs whose co-inactivation is depleted in in vitro and patient tumors. As this fitness reduction is expected to result in better patient survival, ISLE further refines SL prediction by integrating patients' clinical information. While SL interactions (Kelley & Ideker, 2005; Zhong & Sternberg, 2006; Szappanos et al, 2011b; Jerby-Arnon et al, 2014; Srivas et al, 2016; Wang et al, 2017a; Lee et al, 2018) pinpoint molecular vulnerabilities in tumors that can be targeted (the SL partners of genes that are inactivated in a specific tumor) (Weidle et al, 2011; Szczurek et al, 2013), SR interactions can rescue the cells from such vulnerabilities by actively modifying the interacting rescuers, leading to therapy resistance. The SR interaction thus defines an asymmetric relationship between paired genes and, conversely to an SL interaction, undergoes positive selection in patient tumors as it increases tumor fitness, thus leading to adverse effects on patient survival. Accordingly, we develop an in silico approach to identify SR interactions by tailoring the basic ISLE pipeline presented earlier to capture these specific SR features.

## Results

### The INCISOR pipeline and the resulting cancer SR networks

As drugs mainly inhibit target genes, we focus here on two types of SR interactions (Fig 1A): (i) DD-SR (suppressor) interactions, where the Downregulation of a vulnerable gene is rescued by the Downregulation of a rescuer gene (James et al, 1989; Nonet & Young, 1989; Motter et al, 2008; Szamecz et al, 2014; van Leeuwen et al, 2016); and (ii) DU-SR interactions, where the Downregulation of a vulnerable gene is rescued by the Upregulation of a rescuer gene (Sun et al, 2014; Bertotti et al, 2015; Fong et al, 2015; Hugo et al, 2015; Miyamoto et al, 2015; Rathert et al, 2015; Stuhlmiller et al, 2015).

To predict SR interactions, we tailored the statistical tests used in ISLE (Lee et al, 2018) to devise an in silico approach termed "IdeNtification of ClinIcal Synthetic Rescues in cancer" (INCISOR), which is specifically geared to identify SR interactions. Broadly, INCISOR combines multiple lines of evidence—experimental, tumor transcriptomics, survival information, and gene phylogeny—to ascertain whether a gene pair is likely to be SR. Here, we describe the specific steps of INCISOR for predicting DU-SR interactions, where the rescue event is mediated by over-expression (DD-SR prediction follows an analogous approach, Materials and Methods, and Appendix 2 and Fig S1G). INCISOR analyzes in vitro screens and evaluates the extent to which gene phylogeny, molecular, and survival data of patient tumor support the screens. It selects the clinically relevant SR pairs that are supported by all four lines of evidence outlined below. The specific order in which the following four steps are applied sequentially in INCISOR was chosen to minimize the computational cost (Fig 1B, see Materials and Methods for details), as follows:

1  In vitro essentiality screens: This step tailors a recent approach (Wang et al, 2017a) to mine in vitro genome-wide shRNA (Cheung et al, 2011; Marcotte et al, 2012, 2016; Cowley et al, 2014) and drug response screens (Barretina et al, 2012; Iorio et al, 2016) composed of 2.3 million measurements in 720 cancer cell lines. INCISOR analyzes candidate SR pairs in cell lines with a given gene knockdown and identifies the genes whose upregulation is associated with increased cell growth. We term the first gene a vulnerable (V) gene and the second a (DU) rescuer (R) gene. To determine this association between V and R while controlling for cancer types of cell lines used in the screens, INCISOR uses a linear mixed-effects (preprint: Bates et al, 2014) model (see Materials and Methods for details). P-values of association were determined using ANOVA and corrected for multiple hypotheses tested.

2  Molecular survival of the fittest (SoF): By analyzing TCGA gene expression and somatic copy number alterations (SCNA) of 8,749 patients across 28 cancer types, INCISOR selects candidate SR pairs from step 1 that are observed in their rescued state (gene R is specifically upregulated when gene V is inactive) significantly more than expected. This enrichment testifies to a positive selection of samples in the rescued state, a key property of SR interactions. P-value of enrichment was corrected for multiple hypotheses tested.

3  Patient survival screening: Analogous to ISLE, this step further selects those candidate SRs whose rescued state in TCGA tumor samples exhibits worse patient's survival, as the reduced survival can serve as an indicator of increased tumor fitness. INCISOR uses a stratified Cox proportional hazard model to establish this relationship. We systematically control for confounding factors including cancer type, sex, age, genomic instability, tumor purity (Aran et al, 2015), and ethnicity in the Cox model (Materials and Methods).

4  Phylogenetic screening: Because functionally interacting genes are known to co-evolve (Srivas et al, 2016) in a species, we select SR pairs composed of genes with high phylogenetic similarity. The top 5% of phylogenetically similar pairs among the ones passing the previous steps are chosen as the final set of putative SR pairs.

The resulting DU-SR network, which is composed of all the pairwise interactions that pass all four steps described above, is scale-free (Fig 1C, Dataset Table EV2 and EV3) and consists of 1,109

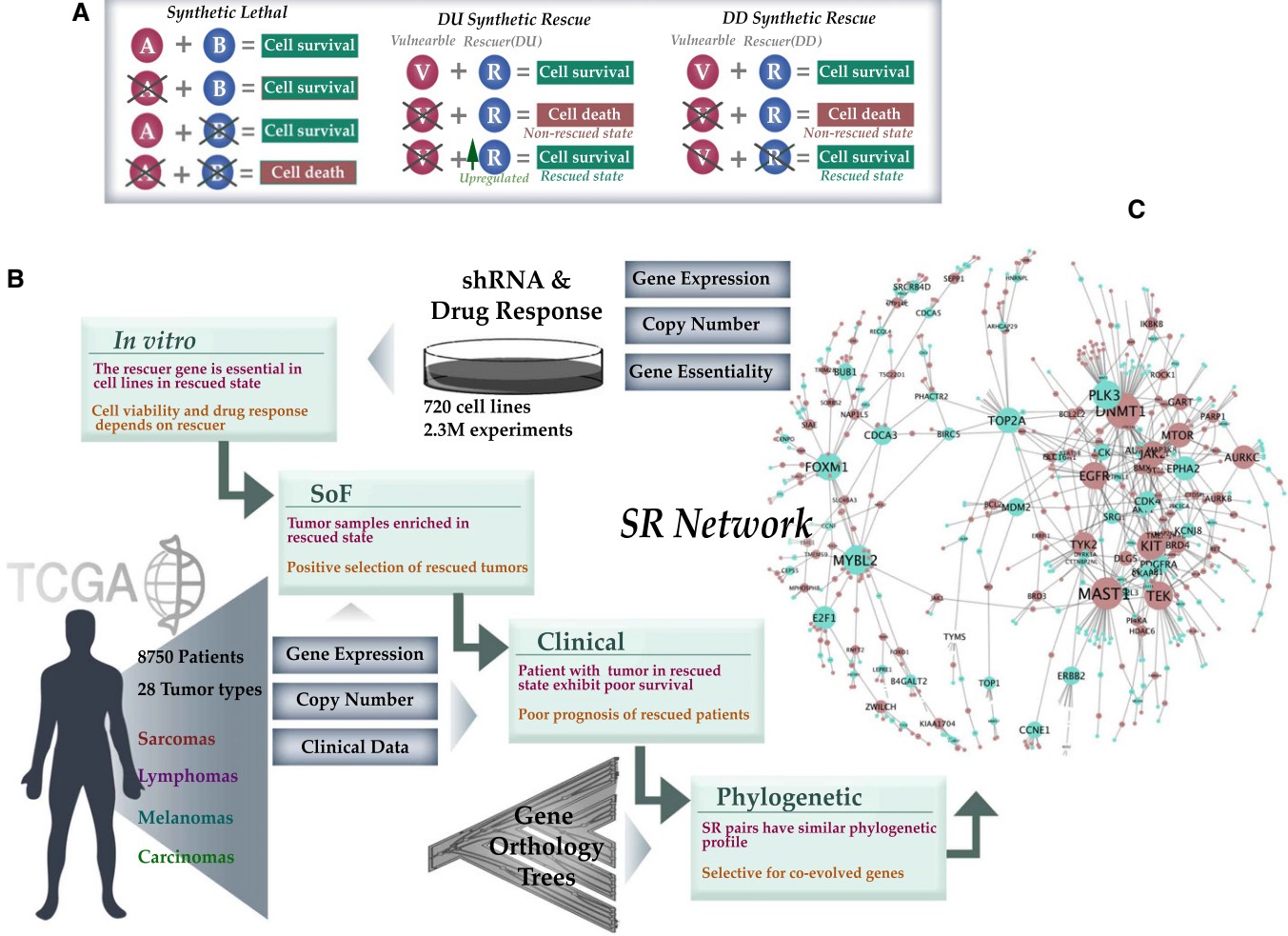

**Figure 1. The INCISOR pipeline and the resulting SR network.**

A   The phenotypic effects of altering interacting gene partners in SL, DD-SR, and DU-SR interactions.

B   The four inference steps of INCISOR and the datasets analyzed (Materials and Methods, SoF stands for the survival of the fittest). The SR property tested (in red) and rationale (in brown) of each step are also displayed.

C   The resulting DU-SR network (purple nodes denote vulnerable genes and green rescuer genes; the size of nodes is proportional to the number of interactions they have). The complete network is provided in Appendix Fig S1F.

genes and 1,033 interactions (see Appendix 2.1 for DD-SR; interactive networks available online, Materials and Methods; Dataset Table EV4 and EV5). Gene enrichment analysis revealed that the network nodes are enriched in cancer and resistance pathways (Appendix 3.5–3.7). We also find that the activation of predicted rescuers increases with advanced cancer stages (Appendix 3.9 and Fig S2G and H). Because cancer type is a major confounder in *in vitro* and patient data, we adopted a statistically rigorous approach to control for cancer type. Age, sex, race, tumor purity, and genomic instability are known to affect patient survival; therefore, we also control for those factors in INCISOR clinical screen in addition to cancer type. We also showed the SR pairs identified are robust to parameter choice in INCISOR (Appendix 2.4). To further check the robustness, we applied INCISOR to breast cancer *in vitro* screens and breast invasive carcinoma (BRCA) patient data from TCGA to identify the breast cancer-specific DU- and DD-SR interactions. The resultant breast cancer SR network is shown to be predictive of breast cancer patients' survival and to a lesser extent, to be predictive of patients' drug response across different cancer types (details in Appendix 3.11).

**Benchmarking INCISOR against a collection of published DU-SR interactions**

We first benchmarked the *DU-SR* predictions via a comparison to genes whose over-expression rescues cancer cells, using a set of genes that were previously shown to mediate cancer drug resistance (Mills *et al*, 2013; Sun *et al*, 2014; Fong *et al*, 2015; Lin *et al*, 2015; Rathert *et al*, 2015; Stuhlmiller *et al*, 2015; Falkenberg *et al*, 2016; Yamaguchi *et al*, 2016; Zhang *et al*, 2016; Dataset Table EV9, See Materials and Methods, Appendix 4.1). INCISOR successfully identified these published rescuer genes with AUCs of 70–85% (mean precision of 46% at 50% recall; Appendix Fig S3O, Materials and Methods). Using a multivariate analysis, we also showed that each

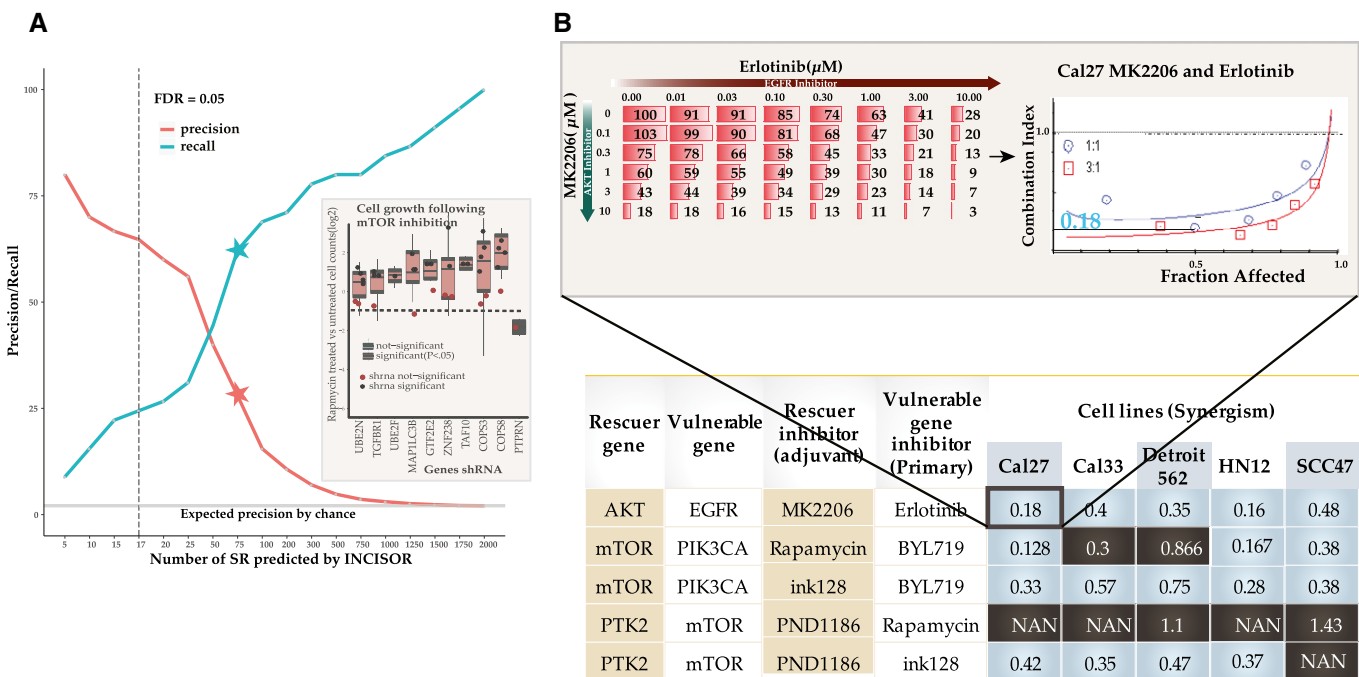

**Figure 2.  Large-scale *in vitro* experiments testing predicted SR interactions in head and neck cancer.**

A   Evaluation of predicted SR (DD) interactions in a large-scale shRNA H&N HN12 cell line screen. The *y*-axis displays the precision and recall of INCISOR-predicted SRs in identifying the 45 experimentally determined *DD-SR* rescuers of mTOR. The vertical dashed line denotes a threshold of FDR = 0.05 over the predicted INCISOR interaction scores. The stars indicate precision and recall at a threshold level where INCISOR identifies 75 genes as *DD-SR* rescuers. The horizontal line (in gray) shows the precision expected by the random chance. The inset displays top 10 predicted genes whose knockdowns are rescued by mTOR inhibition. Significance was quantified using a one-sided Wilcoxon rank-sum test over three technical replicates with at least two independent shRNAs knockdowns per each gene. For 8 of these KDs, at least two shRNA individually show the rescue effect. The black horizontal line indicates the median effect of Rapamycin treatment in controls as a reference point. Box plot limits (Q1, Q3) and whiskers (±1.5 * inter quartile range from hinge) follow a standard definition.

B   Experimental validation of predicted synergistic SR-based combinational therapies in head and neck cancer: A table summarizing the experimentally observed synergism between primary drugs and their predicted rescuer-targeting treatments in 5 HNSC cell lines, based on drug treatment experiments. Synergism was estimated using standard Fa-CI analysis. The table displays the average combination index (CI; synergism CI < 1, additivity effect CI = 1, antagonism CI > 1, NAN indeterminate CI) at 50% growth inhibition (fraction affected). Combinations that are synergistic are colored blue (black otherwise) for each cell lines tested. The inset shows an example of CI calculation for BYL719 and dasatinib combination in HN12 cell lines based on the corresponding dose matrix (number indicates % cell viability at 48 h, n = 3) and Fa-CI curve.

---

screening step of INCISOR contributes to the overall predictive power (Fig S3P and Appendix 4.1). We additionally tested and successfully validated predicted SR interactions using published data of patient-derived *in vitro* (Crystal *et al*, 2014) and mouse xenograft models (Gao *et al*, 2015; Appendix 4.4 and 4.5). As large cohorts of published rescue interactions are still quite scarce, we conducted four new *in vitro* experiments to further test emerging rescue predictions of INCISOR of interest.

**Experimental testing of predicted DD-SR interactions of mTOR in head and neck cancer cell lines**

Our first experiment tested *DD-SR* interactions involving mTOR, a key growth regulating kinase in head and neck cancer. To test the predicted rescue interactions involving mTOR, we knocked down (KD) genes in an experimental screen in a head and neck cancer cell line (HN12) and experimentally identified the (DD) rescue events occurring due to a subsequent mTOR inhibition by rapamycin treatment (which is known to specifically targets mTOR in its complex 1; Laplante & Sabatini, 2012). Because kinases are the most frequent intracellular drug targets, we used a kinase and phosphatase

targeted library for performing knockdowns of 2,214 genes bearing their translational relevance. Forty-five of these KDs, about 2.1%, were rescued by mTOR inhibition in the screen (Dataset Table EV10, Materials and Methods). Independently, we applied INCISOR to identify genes that are predicted to be rescued by mTOR inhibition in a statistically significant manner (FDR = 0.05,). INCISOR predicted 17 such *DD* rescuer genes (Materials and Methods), 11 of which indeed overlapped with the 45 interactions identified experimentally (Appendix Fig S5b). This yields a precision level of ~65% and recall of ~25% (Fig 2A, false positive rate < 0.003), a 31-fold increase over the 2.1% precision expected by chance. INCISOR exhibits a reasonable precision also at high recall rates, e.g., at a threshold INCISOR predicts 75 genes as positive (recall of about 70%), it achieves a precision level of 30% (vs. 2.1% that is expected by random). The validated rescuers were enriched with transcription factors, FoxO signaling and stress response genes (Dataset Table EV30). We further validated the predicted DD-SR interactions of mTOR via multiple published *in vitro* shRNA (Cheung *et al*, 2011; Marcotte *et al*, 2012, 2016; Cowley *et al*, 2014) and drug response screens (Barretina *et al*, 2012; Iorio *et al*, 2016) (Appendix 4.2 and 4.3). In sum, this analysis shows that INCISOR

successfully predicts genetic interactions (of mTOR) whose *functional activation* in cancer cells increases cellular fitness.

**Experimental testing of predicted DU-SR rescuers via drug combinations and siRNA in head and neck cancer**

In the second experimental validation, we tested the ability of predicted *DU-SRs* to guide new synergistic drug combinations, where the combination of drugs hits both a primary cancer drug target and its predicted *DU* rescuer (Materials and Methods). We tested seven such predicted combinational therapies across five different head and neck cancer cell lines. We find that 5 out of 7 combinations are indeed synergistic (Fig 2b, Appendix 4.7, refer to Appendix Fig S6 and S7 for results of all 7 combinations tested). One validated pair involves PI3KCA and mTOR, which are important genes in the PI3K/AKT/mTOR pathway. PIK3CA activates AKT by converting PIP2 to PIP3 (Myers & Cantley, 2010), promoting cell growth and survival. mTOR also promotes cell growth and mTORC2 is known to regulate AKT independent of PIK3CA (Laplante & Sabatini, 2012; Populo *et al*, 2012), thus might compensate for PIK3CA inhibition and explain their synergism.

In the third experiment, we conducted siRNA experiments to show that observations of Fig 2b are consistent. Targeting mTOR by siRNA exhibited enhanced sensitivity to BYL719 in 4 of these cell lines (Appendix 4.7 and Fig S8). Similarly, siRNA targeting of PIK3CA exhibited enhanced dasatinib sensitivity (Appendix 4.7 and Fig S8).

Because many of these drugs tested above are known to target multiple genes, we conducted additional experiments in NSCLC to confirm the relationship between synergism and predicted SRs.

**Targeting predicted DU-SR rescuers of DNMT1 sensitizes resistant NSCLC cell lines to DNMT1 inhibitor**

In the fourth and final *in vitro* experiment, we tested whether targeting predicted *DU* rescuers could sensitize therapy-resistant tumor cells. We picked DNMT1 to test this hypothesis as it is a major hub in the DU-SR network (Fig 1c) and a key cancer gene in non-small-cell lung cancers (NSCLCs). We studied 18 NSCLC cell lines (Materials and Methods) that are insensitive to Decitabine (a DNMT1 inhibitor). In each of these cell lines, we pharmacologically inhibited the 13 top predicted DU rescuers of DNMT1. A Bliss (Bliss, 1939; Lehar *et al*, 2007; Friedman *et al*, 2015) independence model was used to estimate synergism, and its significance was determined by comparing expected vs. observed drug response of drug combinations across all doses tested (Materials and Methods). Targeting the predicted rescuers synergistically sensitized these cell lines to Decitabine in 71% of the 234 (13 rescuers × 18 cell lines) conditions tested. In contrast, pharmacologically inhibition of two top predicted DD rescuers of DNMT1 showed the opposite, *antagonistic* effects, in 64% of the 36 conditions tested, with no synergistic effects, as expected (Fig 3A). Both the observed synergistic and antagonistic effects across cell lines were significantly compared to control drug tested ($P < 2.2E-16$). We further confirmed the ability of predicted SR interactions to predict resistant tumor sensitization in a large published patient-derived cell line collection (Friedman *et al*, 2015) and mice xenograft (Gao *et al*, 2015; Appendix 5.3 and 5.4).

The effects of some of the SR interactions validated in drug combination screen described above can be explained by their known biology. For example, (i) first, DNMT1 epigenetically silences E-cadherin (Robert *et al*, 2003). The silencing results in B-catenin accumulation in cell nucleus (Hayashida *et al*, 2005) that is necessary for maintaining cancer cell stemness. WNT signaling, however, was shown to regulate B-catenin (Colletti *et al*, 2009) independently, explains why WNT1 activation rescues DNMT1 inhibition (Fig 3B). (ii) Second, DNMT1 also silences RASSFA1, which in turns stabilizes the proto-oncogene MDM2 (Zhang *et al*, 2013). Thus, concomitant over-expression of MDM2 could compensate for the loss of RASSFA1 due to DNMT1 inhibition. (iii) Third, CDK1 over-expression may compensate DNMT1 inhibition because CDK1 is known to stabilize DNMT1 by phosphorylating it (Liu *et al*, 2016). (iv) Finally, PAK1 may compensate for DNMT1 inhibition because it independently regulates cell adhesion and motility. These results testify that some rescue interactions may be explained by molecular interactions between genes proximally located on signaling pathways (Kafri *et al*, 2005, 2009; Fig 3B). However, many of the emerging rescue interactions are not, either due to our limited knowledge of signaling pathways or due to functional interactions that go beyond the scope of the signaling pathways.

**Rescuer and vulnerable genes share functional annotations**

Our observation that signaling architecture may explain a subset of SR interactions led to the hypothesis that rescuer and vulnerable genes of SR networks may share functional similarities. Several lines of evidence support this hypothesis. First, in *the DU-SR* network, gene ontology (GO) annotations of rescuers are similar to GO annotations of their partners (Fig 3C). The GO similarity observed in the DU-SR network is significantly higher compared to (i) GO similarity in a random network ($P < 1E-34$) with similar degree distribution as the DU-SR network, and (ii) GO similarity in a network generated by randomly shuffling the interactions between gene pairs of the DU-SR network ($P < 1E-10$). Second, *DU-SR* rescuer genes are significantly closer ($P < 1E-46$ and $P < 3E-10$ compared to the random network and the shuffled network) to their predicted partners in the human protein interaction (PPI) network (Schaefer *et al*, 2012; Fig 3D). Notably, *DU-SR* interactions mediated by direct (physical) protein interactions are enriched in cancer drivers (Fisher's exact test $P < 6.5E-8$, Appendix 3.4). Third, using the STRING database (Szklarczyk *et al*, 2015), which integrates multiple resources of direct and indirect associations of protein interactions, we find that partner genes of the *DU-SR* network are more likely to be functionally related (Fig 3E): Rescuer genes are significantly closer ($P < 5E-72$ and $P < 7E-13$ compared to the random network and the shuffled network) to their predicted partner gene in the STRING network. Moreover, the observed functional similarities between *DU-SR* pairs are not merely due to co-expression between gene partners; shuffled DU-SR gene pairs with similar co-expression levels as those of predicted DU-SR pairs exhibit significantly less GO similarity ($P < 5E-05$). An analogous functional similarity was also observed for gene pairs in the *DD-SR* network (Appendix 3.8 and Fig S2E).

**SR interactions predict drug response in patients**

We next evaluated INCISOR's ability to predict response of patients to cancer drug treatments (Ein-Dor *et al*, 2005; Domany, 2014) by analyzing the transcriptomics of their *pre-treated* tumor samples. To

this end, we applied INCISOR to identify the rescuers of (the targets of) 28 FDA-approved cancer drugs (for which treatment response data are available in the TCGA collection). To remove any potential circularity, during the identification of SR

interactions of targets of a given drug, we removed from TCGA patients who were administered with that drug (Materials and Methods, Appendix Fig S10e). To predict the response of an individual patient's to a given drug, we defined the *drug-tumor SR*

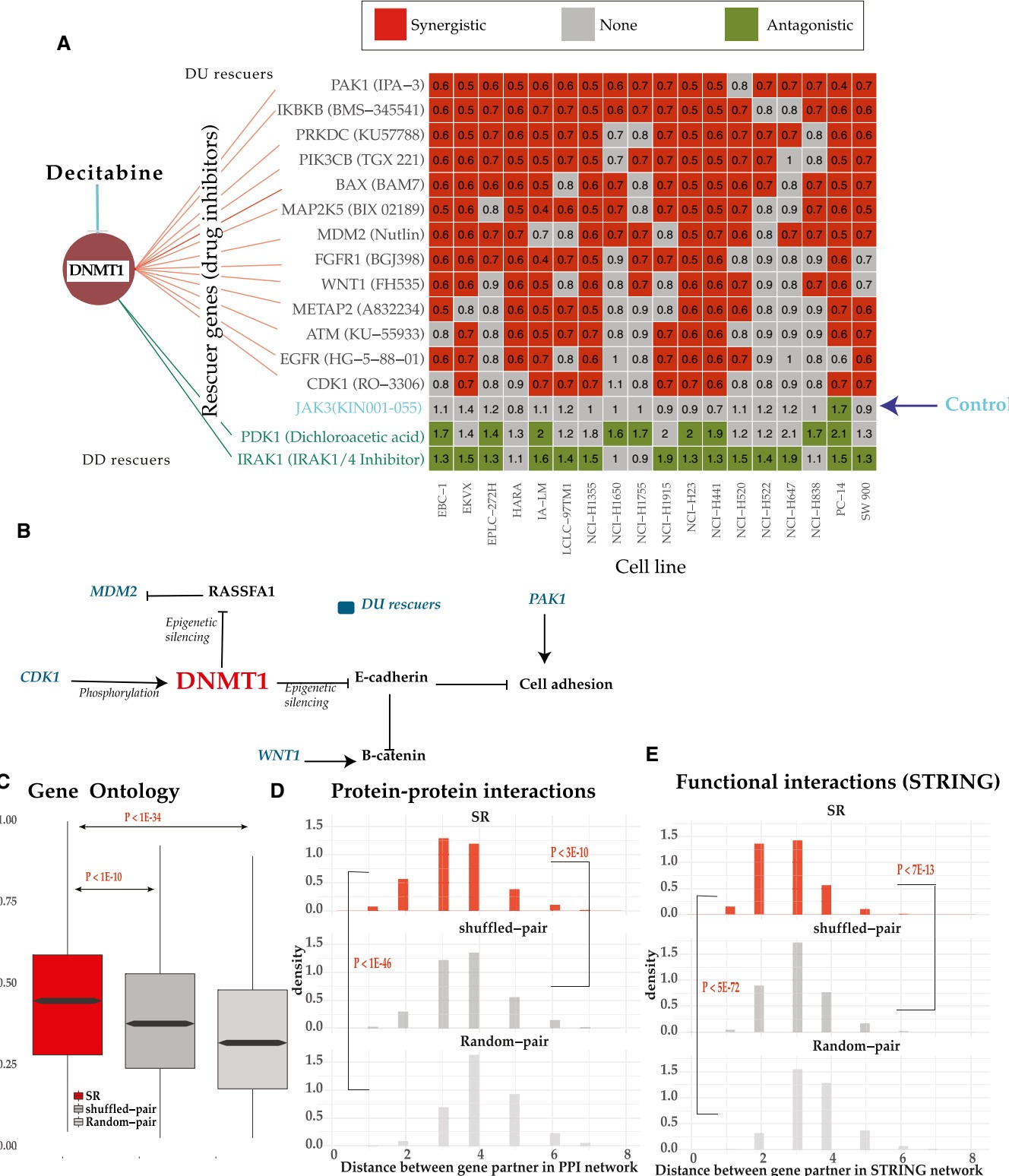

**Figure 3.**

**Figure 3.  Large-scale experiments testing predicted SRs in NSCLC and studying their functional similarity.**

A   Experimental testing of the predicted SR (DU) rescuers of DNMT1 via drug combination experiments in 18 NSCLC cell lines insensitive to Decitabine. The matrix displays drug interactions between Decitabine, a DNMT1 inhibitor, and inhibitors of its predicted rescuer genes (*x*-axis) across 18 NSCLC cell lines (*y*-axis) that are insensitive to Decitabine. Row labels present rescuer genes and their inhibitors. Colors in the matrix show whether the interactions found are significantly synergistic (red), antagonistic (green), or non-significant (in gray). Values in the matrix show average synergism (< 1 synergism and > 1 antagonism, Materials and Methods). Thirteen predicted DU-SR rescuers (red lines), two predicted DD-SR rescuers (green lines) of DNMT1, and one random control (JAK3i) were tested.

B   Some SR interactions of DNMT1 occur between genes proximally located on the signaling pathway. DU rescuer genes of DNMT1 are colored blue.

C–E Functional similarities between gene pairs in the *DU-SR* network. Comparison of functional similarities between interactions in (i) the *DU-SR* network (ii) random pairs (the network is generated by random pairing between protein-coding genes, having a degree distribution similar to that of the *DU-SR* network) (iii) shuffled pairs (the network is generated by shuffling pairing of the DU-SR network). Functional similarities of genes in each pair were evaluated in terms of their (C) GO similarity. Box plots follow standard limits (Q1, Q3) and whiskers (±1.5 * inter quartile range from hinge) follow a standard definition. (D) distances in the human PPI network (Schaefer *et al*, 2012), and (E) distances in the STRING network (Szklarczyk *et al*, 2015). The distances denote the number of interactions on the shortest path between the paired genes. The histogram of network distances between gene pairs is displayed for the PPI and STRING networks. One-sided Wilcoxon rank-sum test was used for significance.

*score* as the number of upregulated rescuers of the drug's targets in that patient's tumor (Materials and Methods). We reasoned that a drug is expected to be less effective in tumors where many of its DU rescuers are upregulated. Using a Cox model to control for confounding variables (Materials and Methods), we find that the SR scores predict the patients' survival after treatment in a statistically significant manner for 22 of the 28 drugs tested (Fig 4A shows the result for the 26 drugs tested with hazard ratios > 1, Materials and Methods). Evaluating the patients' response in terms of tumor size (based on the RECIST criteria), we find that the non-responders exhibit significantly higher drug-SR rescue scores than the responders for 14 out of 19 drugs for which tumor size information was available (Fig 4B, Materials and Methods). An analysis of independent (non-TCGA) ovarian (Patch *et al*, 2015) and breast cancer datasets (Hatzis *et al*, 2011) further shows that SRs

successfully predict both primary and acquired therapy resistance (Appendix 5.2). In contrast, a randomly shuffled network (generated by randomly shuffling rescuer genes for each drug target, maintaining the original SR node degree) fails to predict patients' response to any of the drugs tested, both in the survival-based and response-based analyses. Drug-SLs inferred from DAISY (Jerby-Arnon *et al*, 2014) also showed no predictive signal here (log rank *P* = 0.49). Please note that INCISOR can only predict response to drugs whose gene targets are known.

**Comparative evaluation of INCISOR's performance in predicting drug response vs. other recent large-scale genomic methods**

We compared the performance of INCISOR with other existing methods for predicting cancer drug response. Iorio *et al* (2016)

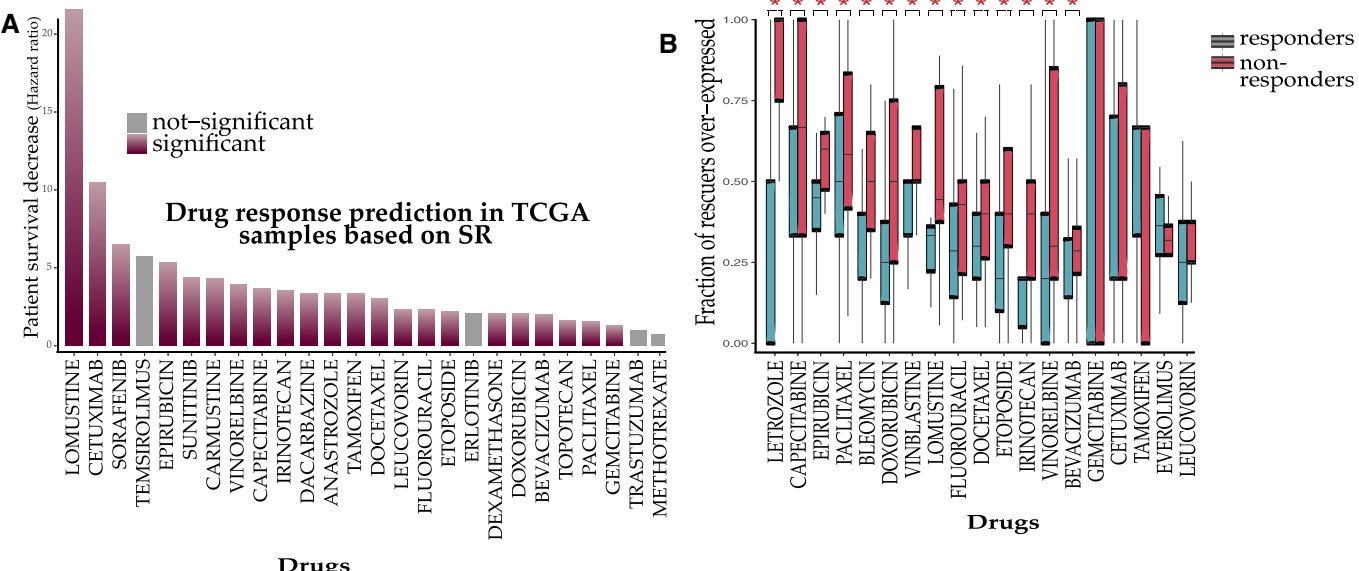

**Figure 4.  SR networks predict cancer drug response in patients.**

A   Prediction of drug response in terms of survival: The *y*-axis displays the hazard ratio of patients as a function of upregulation of predicted rescuers (Materials and Methods).

B   Analyzing drug response in terms of tumor size reduction (RECIST criteria): The predicted *DU-SR* rescuers of drugs are differentially over-expressed in non-responding tumors. The *y*-axis denotes the fraction of the predicted drug-specific rescuers that are over-expressed (out of all predicted rescuers of that drug) in tumors of responders (red) and non-responders (blue). Significant results are marked by stars (Wilcoxon rank sum *P* < 0.05, aggregate Wilcoxon rank sum is *P* < 2.2E-16, Materials and Methods). Box plots follow standard limits (Q1, Q3) and whiskers (±1.5 * inter quartile range from hinge) follow a standard definition.

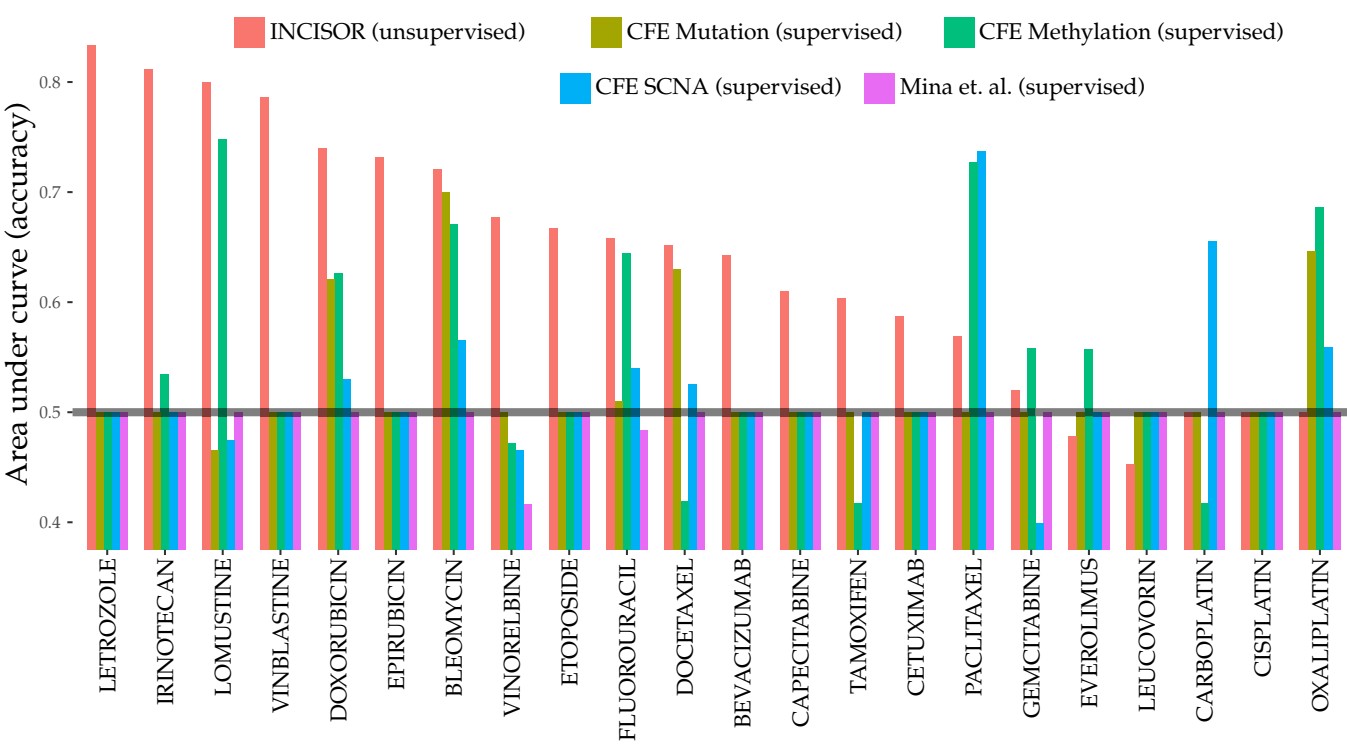

**Figure 5.  Comparative analysis of INCISOR.**

A comparative study of INCISOR's performance (red bars) in predicting patients drug response (TCGA) compared to ISLE- and CFE-based approaches (other colors). The area under the curve (*y*-axis) displays the predictive performance of different methods for 22 FDA-approved drugs in TCGA. Predictions of CFE (cancer functional events) identified by Iorio *et al* (2016) are displayed separately for CFEs inferred from mutation, methylation, and SCNA data.

identified cancer functional events (CFEs) and demonstrated that they could be used to predict drug response of 265 drugs *in vitro*. Similarly, Mina *et al* (2017) identified genetic interactions involving these CFEs and demonstrated they predict drug response in cell lines. To systematically evaluate whether these could also determine drug response in patients, we used the occurrence of CFEs and CFE interactions in patients' tumor as features to build supervised models (Materials and Methods) predicting the response for each drug in TCGA. We analyzed 22 FDA-approved cancer drugs in TCGA, including 19 targeted drugs shown in Fig 4B and three drugs without known gene targets (Carboplatin, Cisplatin, and Oxaliplatin). We also ran the ISLE pipeline (Lee *et al*, 2018) to predict the response to these drugs for further comparison. As shown in Fig 5, while ISLE predicts response for four drugs more accurately compared to INCISOR, INCISOR exhibits better predictive power for 15 drugs as compared to ISLE. The CFEs (of Iorio *et al*, 2016) significantly predict drug response for eight of the drugs (which also includes two non-targeted therapies drugs). However, the CFE-related genetic interactions identified by Mina *et al* do not have a predictive signal for any of these drugs in the TCGA cohort. INCISOR, in turn, outperforms these other methods for 14 FDA drugs (Fig 5). This superiority of INCISOR performance versus that of the CFE-based classifiers is especially notable as it is not based on any supervised training on specific drug response training data and is based on the interactions inferred solely from *pre-treated samples*. Indeed, ISLE predictive performance can be further increased by

about 10% more by building supervised predictors based on INCISOR-predicted rescuers (Appendix Fig S10J).

## SR interactions determine efficacy of immune checkpoint blockades in patients

Finally, we hypothesized that SR-mediated transcriptomic changes mediate resistance to immune checkpoint blockade (ICB; Taylor-Weiner *et al*, 2016). Accordingly, we evaluated INCISOR's ability to predict SRs that can account for key transcriptomic changes occurring in patients' tumors following checkpoint immunotherapy. We also studied the match between the rescuers predicted and key resistance modulators identified in mouse studies. To predict the SR rescuers of the checkpoint genes, we removed *in vitro* essentiality screens (step 1) from the INCISOR pipeline as they are conducted in *in vitro* systems lacking an immune component (Materials and Methods). We find that the pre-treatment expression levels of rescuers of PD1 successfully predict resistance to PD1 blockade in melanoma patients (Fig 6A; Hugo *et al* 2016 and Prat *et al*, 2017, and Appendix 5.5). Similarly, the pre-treatment expression of the INCISOR-predicted rescuers of CTLA4 successfully predicts patients' resistance to CTLA4 blockade in melanoma patients (Fig 6A, Van Allen *et al*, 2015).

To further study the role of SRs in immunotherapy, we consented 40 patients with metastatic melanoma in ongoing clinical trials for treatment with different ICB therapies and carried

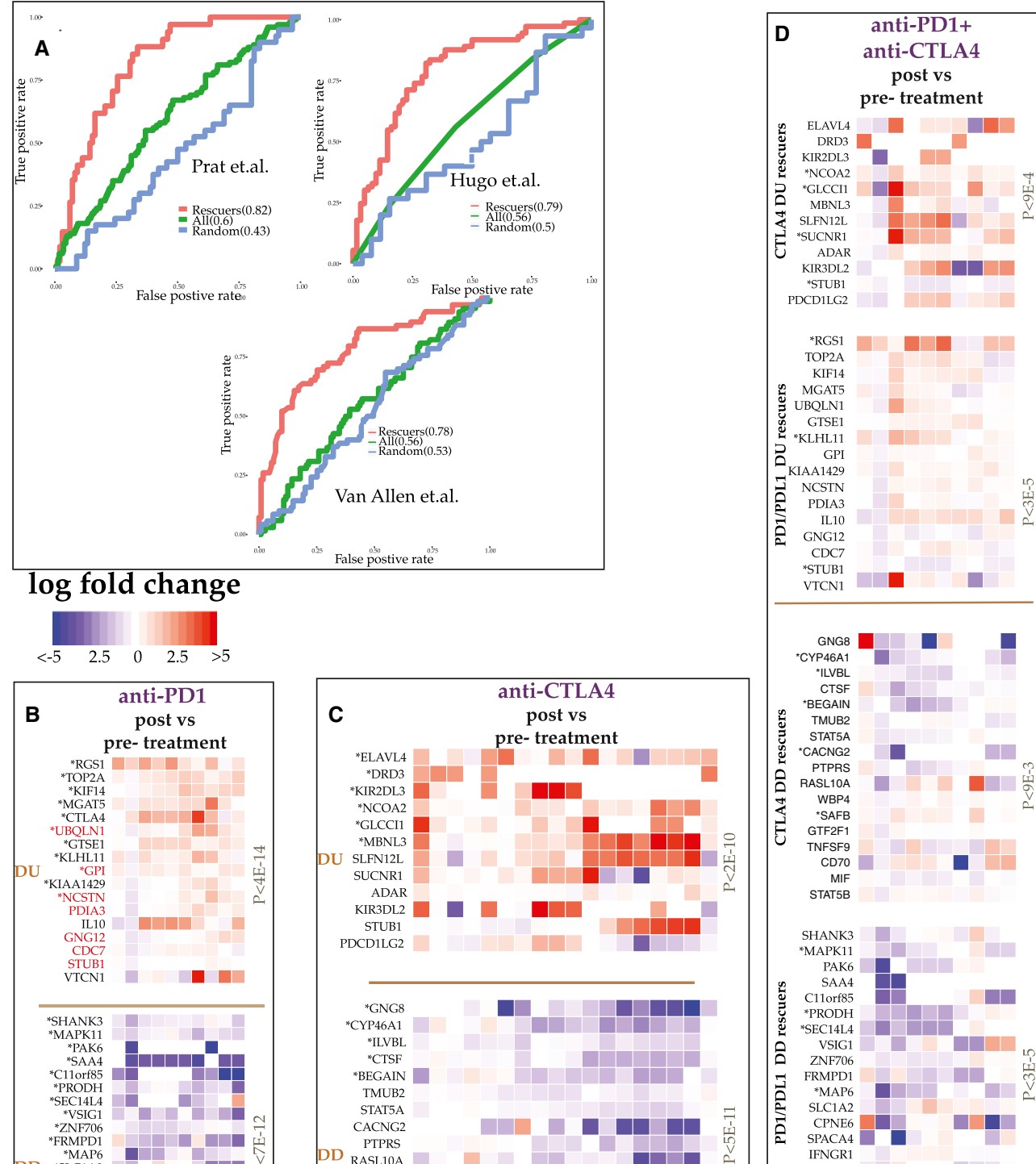

**Figure 6.**

**Figure 6.  SR predicts resistance to PD1/PDL1 and CTLA4 blockade in patients.**

A    Cross-validation accuracy of SR-based supervised predictors in predicting resistance to PD1 (Hugo *et al*, 2016; Prat *et al*, 2017) and CTLA4 blockade (Van Allen *et al*, 2015), reported in terms of the corresponding receiver operating characteristic (ROC) curves. Expression of the predicted rescuer genes of PD1 (CTLA4) was used to train an SVM supervised predictor of PD1 (CTLA4) blockade. For comparison, we also display the ROC curves of supervised predictors trained on the expression of all genes (shown as "All") and on the expression of genes selected randomly and controlled for the number of rescuers predicted (shown as "Random").

B–D  The transcriptomic alterations of rescuer genes post-PD1/PDL1 and CTLA4 blockade in patient tumor biopsies: Their post- (vs. pre-)-treatment expression changes of DU/DD rescuers after anti-PD1 (B), anti-CTLA4 (C), and PD1 + CTLA4 combination therapies (D). Each panel displays the expression fold change of each predicted rescuer gene (rows) for different tumor samples (columns) and the *P*-value of overall paired Wilcoxon test of the expression changes observed in paired samples. Significantly altered up/downregulated genes are marked by (*). Genes marked in red are those whose CRISPR knockdown enhances melanoma sensitivity to anti-PD1 blockade in mice models.

out whole transcriptomics profiling of their 90 matched pre-, on-, and post-treatment tumor biopsies (Materials and Methods, Data available online). Forty biopsies were taken from patients treated with anti-PD1/anti-PDL1 (collated together in the analysis and referred as anti-PD1), forty-three biopsies with anti-CTLA4, and seventeen with a combination of anti-CTLA4 and anti-PD1 (patients who sequentially underwent from first ICB regiment to another were also considered for individual analysis of the first ICB). Notably, post-treatment biopsies were performed when the patients stopped responding to the ICB, denoting the emergence of resistance.

We find that the predicted DU (DD) rescuers of anti-PD1 therapy (Materials and Methods) are upregulated (downregulated) in anti-PD1 post-treatment tumor biopsies (paired Wilcoxon $P < 4E$-14; Fig 6B top and bottom panels, their pathway enrichment is provided in Dataset Table EV28). Notably, the knockdown of 7 of the 17 predicted upregulated DU rescuers of anti-PD1 therapy has been recently found to promote melanoma's sensitivity to anti-PD1 blockade in mice models (hypergeometric enrichment of $P < 8E$-17; colored red in Fig 6B) (Manguso *et al*, 2017). Three of 21 of the predicted DD rescuers have also been identified in that study as enhancing resistance, as expected (hypergeometric enrichment of $P < 5.5E$-7). More specifically, our results provide evidence in humans that support the mice findings, that gene inactivation of *IFNGR1, RABEPK,* and *MIF* induces tumors resistant to PD1 blockade and gene inactivation of *PDIA3, STUB1, CDC7, UBQLN1, NCSTN, GNG12,* and *GPI* co-simulates immune response to PD1 blockade in melanoma. Interestingly, we identify *CTLA4* as a DU rescuer of anti-PD1 therapy, supporting the rationale of their combination. Other notable upregulated DU rescuers of PD1 are the immune checkpoint genes *VTCN1* and *TOP2A*. The latter suggest combinations involving DNA topoisomerase inhibitors such as Doxorubicin and Epirubicin with anti-PD1 as a potential combination therapy. Analogously, top predicted DU (DD) rescuers of anti-CTLA4 therapy were upregulated (downregulated) in post-treatment tumor biopsies derived from patients treated with anti-CTLA4 therapy (paired Wilcoxon $P < 5E$-11, Fig 6C; see Dataset Table EV29 for pathways enrichment). Notably, we find that anti-CTLA4 blockade can be DU-rescued by a class of inhibitory checkpoints—Killer-cell immunoglobulin-like receptors (*KIR2DL2* & *KIR3DL3*), which are known to interact with MHC1 and facilitate cell death (Bashirova *et al*, 2006), putting forward the potential benefits of combinations targeting these genes. Analyzing samples of post-treatment combination therapy involving both anti-PD1 and anti-CTLA4, we find that many DU/DD rescuers respond as predicted but their individual response is evidently weaker (Fig 6D).

## Discussion

In summary, INCISOR prioritizes clinically relevant SRs by analyzing functional genomic and clinical survival data in an integrated manner. Due to the scarcity of published gold standards of SR interactions, we conducted new large-scale *in vitro* experiments to validate our predictions. The paucity of known rescue interactions in the literature further underscores the importance of developing tools like INCISOR. Overall, INCISOR attained precision levels of an average 48% (at 50% recall) in the identification of true SR interaction across all published and new experiments. Finally, we show that SR mediates both primary and adaptive resistance in patients: e.g., we show that the pre-treatment expression data of TCGA tumors are predictive of their response to drug treatments (primary resistance, Fig 4), and on the other hand, SRs can predict the post-treatment alterations following checkpoint inhibitors (adaptive resistance, Fig 6B–D).

Like many genome-wide approaches, INCISOR has several limitations, including pitfalls arising from gene co-expression and from correlations in the copy number alterations of proximal genes, which may lead to the inference of false positive SRs. We have verified that the SR interactions are not biased toward genes lying on the same chromosome (Appendix 2.3). We aimed to mitigate false positives in the design of INCISOR by selecting candidate SR pairs only when they are additionally supported by the shRNA and phylogenetic data that testify to causal rescue effects. Although INCISOR explains molecular mechanism of resistance to targeted therapies, it fails to capture resistance mechanism of untargeted therapies. Further, for many drugs, resistance can emerge via mechanisms independent of SRs, e.g., resistance due to alteration in drug efflux. We experimentally validated many of predicted SR interactions using gene inhibition by shRNA and drug treatment screens. To validate the rescue effect due to mTOR inhibition, we used rapamycin that blocks preferentially mTOR in its mTORC1 complex. To discount the possibility that observed synergism is not due to non-specific targeting of drugs, we conducted a large-scale drug combination screen. However, these findings must be further confirmed by CRISPR experiments to completely eliminate the possibility of off-target effects as confounders. Using a multivariate logistic regression, we also show that each screening step of INCISOR contributes to its overall predictive power. These contributions vary across different datasets, testifying that combining the screens is a good strategy and with the growing availability of validated SR interactions, the performance of INCISOR could be improved in the future by adopting a supervised strategy. We expect a higher false positive rate from INCISOR when predicting SRs of ICB treatments because the first screening step cannot be performed. Finally, as this is the

first genome-wide study of cancer SR interactions, we focused on identifying SRs that are common across many cancer types. INCISOR identifies the same interactions for all drugs targeting the same gene(s). Future studies, however, will further identify cancer-type-specific or context-specific SR networks as more data accumulate.

Multiple rescuer genes could rescue and cause resistance to a given cancer drug. Three different strategies could be adapted to prioritize gene target among such multiple rescuers to maximize their clinical benefit: First, INCISOR quantifies that extent of the rescue for each rescuer based on its clinical significance observed in patients. This could be used for prioritization. Second, post-treatment transcriptomic data from a patient's tumor, if available, could be used to narrow down rescuer alterations specific to that tumor. Finally, combining experimental testing in a patient's tumor material using organoids or PDXs with INCISOR predictions would be a powerful approach to systematically identify true clinically relevant rescuer among the multiple predicted SR rescuers.

This study has focused on the genome-wide prediction of SR interactions. Evidently, different signaling functional and physical interactions may be manifested in these rescue interactions (Fig 3B–E). SRs are much less known and studied compared to another type of genetic interactions, known as synthetic lethal (SL) interactions. The difference between SL interactions and *DD-SR* interactions is obvious, by definition. Their difference from *DU-SR* interactions is more intricate: It manifests itself in cells where a given gene is in its wild-type state and its partner interacting gene is knocked down; if the two genes SL interact, there will be no reduction in cellular fitness in that case, but if they DU-SR interact, then the knockdown will reduce cellular fitness (as the rescuer is not upregulated). Consequently, our results demonstrate that a given cancer drug may be effective in cells where its predicted rescuer is in its wild-type state but may become resistant as it is over-expressed. As expected, as SL interactions predict patient-specific primary vulnerabilities, while SR interactions predict a therapy resistance overcoming such vulnerabilities, we found no overlap between predicted DU-SR interactions and SL interactions, predicted by a similar data mining approach ISLE (Lee *et al*, 2018; Appendix 2.2). In general, INCISOR identifies fewer genetic interactions as compared to ISLE (Materials and Methods). Importantly, because ISLE and INCISOR capture complimentary landscape of tumor fitness, predictions from both approaches could be combined in future studies. To demonstrate the potential of such an approach, we present, for instance, the results of such a combining SL and SR in predicting patient survival in breast cancer (Appendix Fig S2I and J).

In conclusion, we present a comprehensive approach to tackle resistance to targeted and immune cancer therapy by mining thousands of tumors available in TCGA to infer cancer-specific SR interactions. We conducted *in vitro* experiments demonstrating that targeting predicted DU-SRs could sensitize therapy-resistant tumor cells, identifying synergistic drug combinations. As SR interactions are derived directly from analyzing the patients' clinical samples, they are more likely to be clinically relevant (Raphael *et al*, 2017) than findings based on cell screens and mouse models solely. Our results lay a basis for the development of new combination therapies based on the molecular characteristics of an individual patient's

tumor to proactively overcome resistance in a precision based manner.

# Materials and Methods

### The INCISOR pipeline for identifying SR interactions

INCISOR identifies candidate SR interactions employing four independent statistical screens (Fig 1B), each tailored to test a distinct property of SR pairs. We describe here the identification process for the DU-type SR interactions (Down–Up interactions), where the up-regulation of rescuer genes compensates for the downregulation of a vulnerable gene (e.g., by an inhibitor compound, Appendix Fig S1A). Then, we discuss how to modify DU-INCISOR to detect the other SR types (DD, UD, and UU). We identify pan-cancer SRs (that are common across many cancer types) analyzing gene expression, somatic copy number alteration (SCNA), and patient survival data of TCGA (Weinstein *et al*, 2013) from 8,749 patients in 28 different cancer types. INCISOR also integrates predictions from TCGA data with genome-wide shRNA (Cheung *et al*, 2011; Marcotte *et al*, 2012, 2016) and drug response (Barretina *et al*, 2012; Iorio *et al*, 2016) screens in around 720 cell lines composing in the total of 2.3 million shRNA measurements. The same approach can be used to identify cancer-type-specific SRs, in an analogous manner. INCISOR is composed of four sequential steps (an FDR threshold was set 0.05 for each step):

1  *In vitro screening (using in vitro cancer data):* Mining large-scale *in vitro* shRNA and drug response datasets, INCISOR examines all possible gene pairs to identify putative SR. The screen adopts an analogous approach (Wang *et al*, 2017a) to mine shRNA screen in a reference collection of cell line to identify pairs where vulnerable genes V and rescuer genes R fulfill the following two conditions: (i) Knockdown of V exhibits an increase in cell growth in cell lines with R upregulated (relative to cell line with R downregulated), and (ii) knockdown of the R is lethal in cell lines where V downregulated. We use both gene expression and SCNA data to identify such putative SR.

To determine this association between V and R while controlling for cancer types of cell lines used in the screens, INCISOR uses a linear mixed-effects (46) model. *P*-values of association were determined using ANOVA and corrected for multiple hypotheses tested. For each input screen, we model cancer types of cell lines as a random effect in the linear mixed-effects model (46). Specifically, we model the effect of a vulnerable gene knockdown ($y$) on cell proliferation as a linear mixed-effects model of its rescuer expression ($g$) and cancer type, where g is modeled as fixed effect and cancer type is modeled as random effect as follows:

$$y \sim g + (1|cancer\_type)$$

Here, we follow the standard notation $(1|cancer\_type)$ to represent the random effect of the confounding cancer type. In the case of shRNA screens, y represents gene essentiality of the vulnerable and in the case of drug screens y represents the IC50 of a drug that inhibits the vulnerable gene. $y$ is quantile normalized to N(0,1) and parameter is estimated using the lme4

software (46). The *P*-value of the fixed effect was estimated using ANOVA. *P*-values were adjusted for multiple hypotheses by calculating the false discovery rate considering the number of hypotheses (pairs) tested in each screen.

SCNA-based conditional essentiality is determined analogously. Putative SR pairs significant either in shRNA screen or in drug response screen either using gene expression or SCNA are referred as putative SR. We apply the standard FDR correction (Benjamini & Hochberg, 1995) in this step. Specifically, to combine *P*-values from multiple datasets of shRNA and drug screens that were processed, we tested two alternatives: (i) We first calculated the adjusted *P*-value within each dataset and then applied multiple hypothesis correction on the adjusted *P*-value for each pair tested. (ii) We also tried the Fisher's method (Poole *et al*, 2016) to rigorously combine *P*-values across all datasets in step 1 and then applied FDR on Fisher-combined *P*-values. The results from both alternative approaches were identical. Pairs significant either using gene expression or SCNA are referred as putative SR and are passed on to the next screen.

2  *Molecular survival of the fittest (SoF, analyzing tumor molecular data)*: This screen mines gene expression and SCNA data of the input tumor samples to identify vulnerable gene (V) and rescuer gene (R) pairs having the property that tumor samples in the *non-rescued* state (that is, samples with underactive gene V and non-overactive gene R, activity states 1 and 2 in Appendix Fig S1A) are significantly less frequent than expected, whereas samples in the *rescued* state (that is, samples with underactive gene V but overactive gene R) appear significantly more than anticipated (testifying to the positive selection of *rescued* state of the pairwise interaction). The significance of the enrichment/depletion of rescued/non-rescued state is determined via a hypergeometric test followed by standard false discovery rate correction. A gene is defined as inactive (respectively, overactive) if its expression level is less (greater) than the 33rd percentile (67th percentile) across samples for each cancer type (to control for cancer type). Otherwise, it is considered to have a normal activation level. Out of total N tumor samples, if n1 (n2) is the number of samples in the rescued/non-rescued state using specific activation level of gene R (V) independently, k is the number of samples in the activity state using both genes R and V, the significance of enrichment/depletion of the observed number of samples in the rescued/non-rescued state is determined using hypergeometric test: $hypergeometric(k, n1, N, n2)$. Enrichment/depletion of the activity state using SCNA is set analogously. Pairs significant (FDR < 0.05) in both SCNA and mRNA are passed on to the next screen.

3  *Clinical screening (using patient survival data)*: This step selects a gene pair as SR if it has the property that tumor samples in *rescued* state (that is, samples with underactive gene V and overactive gene R) exhibit significantly poorer patient's survival and samples in *non-rescued* state tumors exhibit better survival than rest of the other samples. Specifically, INCISOR uses a stratified Cox proportional hazard model to check such observed associations of SR *rescued*/*non-rescued* state are significantly larger compared to the expected additive survival effect of their individual genes, while controlling for confounding factors including cancer type, sex, age, genomic instability, tumor purity, and race (shown here for expression analysis for an activity state *A* and a similar model is used to analyze SCNA data):

$$h_g(t, patient) \sim h_{0g}(t) \exp(\beta_1 I(V, R) + \beta_2 g(V) + \beta_3 g(R) + \beta_4 age + \beta_5 GII + \beta_6 TP) , \quad (1)$$

where g is a variable over all possible combinations of patients' stratifications based on cancer type, race, and sex. $h_g$ is the hazard function (defined as the risk of death of patients per unit time), and $h_{0g}(t)$ is the baseline hazard function at time t of the $g^{th}$ stratification. The model contains six covariates: (i) $I(V, R)$: indicator variable representing if the patient's tumor is in the activity state *A*, (ii) $g(V)$ and (iii) $g(R)$: gene expression of V and R, (iv) age: age of the patient, (v) GII: genomic instability index of the patient, and (vi) TP: tumor purity. The βs are the unknown regression coefficient parameters of the covariates, which quantify the effect of covariates on the survival.

All covariates are quantile normalized to $N(0, 1)$. The βs are determined by standard likelihood maximization (Andersen & Gill, 1982; Therneau & Grambsch, 2013) of the model using the R-package "Survival". The significance of $\beta_1$, which is the coefficient for the SR interaction term, is determined by comparing the likelihood of the model with the NULL model without the interaction indicator $I(A, B)$ followed by a likelihood ratio test and Wald's test (Andersen & Gill, 1982; Therneau & Grambsch, 2013), i.e.,

$$h_{null}, g(t, patient) \sim h_{0g}(t) \exp(\beta_2 g(V) + \beta_3 g(R) + \beta_4 age + \beta_5 GII + \beta_6 TP) . \quad (2)$$

The *P*-values obtained are corrected for multiple hypothesis testing. We pass a putative SR pair to the next screen if its *rescued state* exhibits significantly poorer survival and the *non-rescued state* exhibits better survival regarding both mRNA and SCNA (all FDR < 0.05).

Tumor purity is obtained for each TCGA sample from Aran *et al* (2015). They combined following four methods to estimate an aggregate estimate of tumor purity: (i) ESTIMATE (Yoshihara *et al*, 2013), (ii) ABSOLUTE (Carter *et al*, 2012), (iii) LUMP, and (iv) IHC. We control for tumor purity estimated from each of these four methods in addition to the combined tumor purity values by (Aran *et al*, 2015) in the survival analysis.

The above modeling of survival as stratified Cox regression allows to account for systematic differences in survival in different cancer types. INCISOR assumes a different baseline hazard for each cancer type to compute likelihoods. The estimated likelihoods are then combined to estimate the effects of gene interactions on survival.

4  *Phylogenetic profiling screening*: We further filter and select SR pairs composed of genes having high phylogenetic similarity, motivated by the findings of Srivas *et al* (2016). This is done by comparing the phylogenetic profiles of the SR-paired genes across a diverse set of 87 divergent eukaryotic species adopting the method of Tabach *et al* (2013a,b). The resulting matrix of the phylogenetic scores of all candidate genes is clustered using a non-negative matrix factorization (NMF; Kim & Park, 2007), and the Euclidian distance between the cluster membership pattern of each gene in given candidate pair is computed. The significant (empirical-FDR < 0.05) phylogenetically similar pairs are predicted as the final set of SR pairs.

Please note SR pairs significant either using gene expression *or* SCNA are referred as putative SR in step 1. Applying the stringent "AND" condition in step 1, that is, requiring that the putative SR pair must be significant in both SCNA and gene expression in *in vitro* datasets, results in the removal of many SR pairs that have been actually reported in the literature. This is likely to be an artifact arising since many *in vitro* datasets are simply missing SCNA information for many of the genes. In difference, the "AND" condition could be applied consistently in steps 2 and 3 because the TCGA collection does have SCNA information on all genes for all the patient samples analyzed.

Dataset Table EV31 provides the number of pairs filtered after each screening step. The first step of INCISOR uses both shRNA and drug response screens to compile putative *in vitro* SRs, which tests 15,486 × 19,001 (shRNA KD) hypotheses. We have adjusted the *P*-value using Benjamini–Hochberg and apply FDR < 0.2 accordingly, to identify 2878319 putative (DU) shRNA screen-derived interactions. In the case of drug response, the hypothesis space is smaller, with 221(number of drugs) × 19001 KDs. We again applied a FDR < 0.2 threshold and identified 354K putative drug-SR interactions. The first step ends up identifying 3 million (DU) SR candidates; the second step identified 1.2 million pairs; and the third step reduced selected 9,021 pairs, followed by 1,033 interactions in the final step.

To process half a billion gene pairs for around 9,000 patient tumor samples in a reasonable time, the most computationally intensive parts of INCISOR are coded in C++ and ported to R. Further, INCISOR uses open Multiprocessing (OpenMP) programming in C++ to use multiprocessor in large clusters. Also, INCISOR performs coarse-grained parallelization using R-packages "parallel" and "foreach". Finally, INCISOR uses Terascale Open-source Resource and QUEue Manager (TORQUE) to uses more than 1,000 cores in the large cluster to efficiently infer genome-wide SR interactions.

**Applying INCISOR to construct the DD-SR network**

*Constructing the DD-SR network*
We modified INCISOR in the DD-SR network inference to account for the fact that rescuer gene downregulation leads to synthetic rescues. In DD, the *rescued* state is defined as co-inactivation of vulnerable (V) and rescuer gene (R); and *non-rescued* state is defined as underactive gene V and active gene R (Appendix Fig S1B). Accordingly, the four screens of INCISOR, described above for DU identification, were modified as follows: (i) SoF and Survival screening: The statistical tests (i.e., hypergeometric test and Cox regression) are modified so as to account for DD interactions that have different activity states (i.e., *rescued* and *not-rescued* states, Appendix Fig S1B). (ii) shRNA screening: Similarly, the *conditional* knockdown of a DD rescuer gene now increases the cell proliferation due to activation of DD synthetic rescue. The significance of the increase in the cell proliferation due to a rescuer downregulation is quantified in an analogous manner using Wilcoxon rank sum test. (iii) Phylogenetic screen: It remains the same as the case of DU identification (refer to Appendix 2 for additional details).

*Interactive SR networks*
The four types of SR networks for pan-cancer were created using Cytoscape (Kraskov *et al*, 2005) and are accessible online in an

interactive manner at http://www.umiacs.umd.edu/~vinash85/private/SR/ (with username: "sr" and password: "sr123").

*Genomic instability index*
Genomic instability index measures the relative amplification or deletion of genes in a tumor based on the SCNA. Given $s_i$ be the absolute of log ratio of SCNA of gene $i$ in a sample relative to normal control, GII of the sample is given as (Bilal *et al*, 2013):

$$GII = 1/N \sum_1^N I(s_i > 1).$$

**Calculation of INCISOR interaction score**

INCISOR evaluates each of the candidate SR gene pairs based on the strength of their SR interactions. We define *INCISOR interaction-score,* which combines the significance levels of the four statistical tests in the INCISOR pipeline. First, for each screen, the statistical significance levels of all gene pairs tested were rank-normalized to a value between 0 and 1 (with 0 representing a pair with the highest significance and 1 with the lowest). The final INCISOR interaction-score for a gene pair $i$ is given as:

$$\text{Interaction score} = r_i(1) + r_i(2) + r_i(3) + r_i(4), \tag{3}$$

where $r_i(k)$ represents the rank normalized value of the $k^{th}$ screen of INCISOR.

**Mapping of drugs to their gene targets**

The drugs were mapped to their targets based on the mapping reported in CCLE, CTRP, and DrugBank (Knox *et al*, 2011; Garnett *et al*, 2012; Iorio *et al*, 2016) with exception of target genes whose mechanism of action is explicitly denoted as an agonist in DrugBank.

**Effect size via Cohen's d**

Throughout the manuscript, whenever applicable, to quantify a difference between two groups, we use an effect-size measure called *Cohen's d* (Cohen, 1992). It is defined as the difference of means divided by pooled standard deviation. Given $s_1$ and $s_2$ as standard deviations of two groups and $n_1$ and $n_2$ are a number of samples in each group, the pooled standard deviation is defined as:

$$\sqrt{\frac{(n_1 - 1)s_1^2 + (n_2 - 1)s_2^2}{n_1 + n_2 - 2}}. \tag{4}$$

**Pathway enrichment**

GO and KEGG enrichment analyses were conducted using R-packages *clusterprofiler* and *GOFunction* using default settings.

**Precision and recall**

Using standard definitions, we define INCISOR's precision as the fraction of true SR interaction among the predicted SR interaction by INCISOR. The INCISOR's recall is defined as the fraction of true SR

interactions that are retrieved by INCISOR among all true SR interactions.

## Benchmarking DU-SR networks using literature compiled SR interactions

The seven datasets of the published SR interactions were compiled using extensive literature survey of large clinical and experimental studies (Datasets, dataset pairs, and associated publications are listed in Dataset Table EV9). Each dataset consists of a drug and experientially and/or clinically validated genes whose over-expression causes resistance to the drug treatment in patient samples/cell lines. In each study, the pairings between the drug targets (vulnerable genes) and the corresponding resistance-causing genes (rescuer genes) form the positive set; and the pairings between the targets and all other genes tested, which do not exhibit resistance, form the negative set. Using the INCISOR *interaction score* of individual SR pairs as the prediction for the strength of SR interaction, we performed standard ROC and precision-recall analysis (Appendix 4.1).

## Constructing the drug-DU-SR network

To remove any potential circularity in drug response prediction, for each drug analyzed, we excluded from TCGA dataset the samples of the patients who were treated with that drug. Next, we applied INCISOR to the remaining TCGA samples to identify rescuers of the targets of the drug. The resultant drug-DU-SR network applied for 28 targeted drugs constitutes 182 rescuer genes of 24 drug targets (Appendix 5.1).

## Predicting pan-cancer drug response in patients

### Prediction of drug response using patient survival

Using the drug-DU-SR network, we analyzed 4,328 TCGA samples, which is the collection of samples of patients who were treated with the drugs that were administered to at least 30 patients in TCGA. We predicted that patient tumors would be resistant to drug treatment if multiple DU rescuer genes of the drug targets are upregulated in their tumor. Therefore, the number of rescuer gene overexpressed will be predictive of patients' drug response. Accordingly, for a drug tested $D$ and each patient administering $D$, we estimate the fraction ($C$) of DU rescuer genes upregulated of its drug targets (deduced from their gene expression and SCNA values in the pretreatment tumor sample) in the patient sample. To predict the response of TCGA patients treated, we evaluated the association of $C$ with the patients' survival using stratified Cox model, which also controls for confounding factors (cancer type, age, sex, and race) as follows:

$$h_g(t, \text{patient}) \sim h_{0g}(t, \text{patient}) \exp(\beta_1 C + \beta_2 \text{age} + \beta_3 \text{GII}), \qquad (5)$$

where $h_g$, $h_{0g}$, $\beta$s *age,* and GII are defined as in the equation (1). Covariates $C$, *age,* and GII are quantiles normalized to $N(0, 1)$. The significance of $\beta_1$, which is the coefficient of $C$, is determined by comparing the likelihood of the model with the NULL Cox model, which is similar to (3) but without the covariates $C$, followed by likelihood ratio and Wald's tests (Andersen & Gill, 1982; Therneau & Grambsch, 2013). As evident, SRs can be successfully used to predict drug response in an unsupervised manner (which is hence less prone to over-fitting).

### Prediction of patient drug response based on post-treatment patient tumor size

We evaluated the performance of our prediction vs. TCGA drug response based on patient tumor size following the treatment. Based on RECIST drug response profile of 3,872 patients in TCGA, which were annotated into complete response (CR), partial response (PR), stable disease (SD), and progressive disease (PD), we divided the samples into responders (CR and PR) vs. non-responders (PD and SD). To determine the ability of SR to predict drug response of each drug, we compared the fraction of the DU rescuers (of the drug's targets) upregulated in patients' tumors ($C$), and their significance is determined using Wilcoxon rank sum test.

## Comparative performance of INCISOR in predicting drug response

### DU-SR-based unsupervised predictor

To predict the response of a drug in an unsupervised manner, we first identified responders and non-responders in TCGA dataset. The fraction of over-expressed rescuers of targets of the drug in each patient was used to estimate the area under the curve (AUC). If AUC > 0.5 and mean fraction of over-expressed rescuers was higher in responders compared to non-responders (1-AUC) was used as the final estimate of AUC.

### Supervised prediction of patient response using CFE (Iorio et al, 2016)

The list of CFEs was collected from Iorio *et al* (2016). It provides three distinct types of CFEs: (i) mutation, (ii) methylation, and (iii) somatic copy number alteration (SCNA). Predictive performance of each type of CFEs was evaluated individually. Using TCGA data, we generated a matrix of CFE occurrence across all TCGA patients. The CFE occurrence matrix was used as features to train supervised models for predicting patients' response for each of 22 FDA-approved drugs as follows.

To predict the response of a drug in a supervised manner, we first identified responders and non-responders in TCGA dataset. Given the CFE occurrence matrix as features described above, we built a random forest-based supervised predictor that discriminates responders from non-responders. The random forest was preferred over SVM because its performance was superior as compared to SVM for this prediction task. Twofold cross-validation was used to estimate AUC.

### Supervised prediction of patient response using CFE interaction Mina et al (2017)

The list of CFEs was collected from Mina *et al* (2017). ANOVA *P*-value < 0.05 was used to filter out non-significant drug and CFE pairs, resulting in 1,444 drug CFE pairs with significant association. CFE occurrence in TCGA patients was downloaded from www.ciriellolab.org/select/select.html and was used to identify whether CFE pairs co-occur in the patient's tumor, which is represented as an indicator variable. To predict the response of a drug, we used CFE pairs reported to significantly associate with the drug by Mina *et al* as features. The corresponding matrix of CFE pairs co-occurrence in

patients was used to train a supervised model in an analogous manner described above for Iorio *et al*.

### Experimental testing of INCISOR-predicted SR interactions involving mTOR

We used rapamycin because it is a mTOR inhibitor and hence enables targeting of a predicted rescuer gene by a specific drug, combined with the ability to knock down predicted vulnerable genes in a clinically relevant laboratory setting. Rapamycin is known to specifically targets mTOR in its complex 1 (Laplante & Sabatini, 2012). Its selectivity stems from the need to act on a protein FKBP12, which binds to the FKBP12-binding region (FRB) in mTOR (Huang *et al*, 2003). This was confirmed in our earlier work (Amornphimoltham *et al*, 2008) (particularly in the HN12 cell line, which we used in our experiment) by expressing an FRB mutant mTOR that cannot bind to the rapamycin-FKBP12 complex, which rescued these cells from the anti-tumor effect of rapamycin *in vitro*. Further, long-term treatment with rapamycin has been shown to inhibit mTORC2 in various cellular systems (Laplante & Sabatini, 2012). Indeed, we have previously shown this to be the case in HNSCC, in which we see evidence of mTORC2 inhibition after 2 days of treatment with rapamycin in numerous HNSCC experimental models (Amornphimoltham *et al*, 2008; Iglesias-Bartolome *et al*, 2012; Martin *et al*, 2014) and after short-term treatment of HNSCC patients with rapamycin (Wang *et al*, 2017b). This retro-inhibition approach further supported the specificity of rapamycin for mTOR, in a biologically relevant context. Therefore, we choose HN12 to conduct this experiment. Kinases are the most frequent intracellular drug targets; therefore, we used a kinase and phosphatase targeted library for knockdown 2,214 kinases. Knockdown efficiency of the kinase shRNA library was validated in our prior studies (Lee *et al*, 2014).

Appendix Fig S5A provides an overview of overall experimental procedure (Appendix 4.6). We performed the shRNA knockout and mTOR inhibition in the following steps. 2,214 gene kinases (Dataset Table EV10) were knocked down in HN12 cell lines. HN12 cells were infected with a library of retroviral barcoded shRNAs at a representation of ~1,000 and a multiplicity of infection (MOI) of ~0.3, including at least two independent shRNAs for each gene of interest and controls. At day three post-infection, cells were selected with puromycin for 3 days (1 μg/ml) to remove the minority of uninfected cells. After that, cells were expanded in culture for 3 days, and then, an initial population-doubling 0 (PD0) sample was taken. For *in vitro* testing, the cells were divided into six populations, three were kept as a control and three were treated with rapamycin (100 nM). Cells were propagated in the presence or not of a drug for an additional 12 doublings before the final PD13 sample was taken. shRNA barcode was PCR-recovered from genomic samples and samples sequenced to calculate the abundance of the different shRNA probes. From these shRNA experiments, we obtained cell counts for each gene knockdown at the following two time points: (i) post-shRNA infection (PD0, referred as initial count), and (ii) shRNA treatment followed by either rapamycin treatment (PD13, referred as treated count, three replicates) or control (PD13, referred as untreated count, three replicates).

Significant experimental (DD) rescue event was determined by using Mageck (Li *et al*, 2014) as follows. The difference of treated and untreated count was modeled as a negative binomial distribution and was used to test whether the difference is significant for each gene tested in pooled shRNA (Li *et al*, 2014). Mageck provides significance and effect size (as log fold change) between treated and untreated conditions for each gene. Forty-five gene knockdowns showed significant rescue effect ($P < 0.05$) after adjusting for multiple hypothesis. Next, we applied INCISOR to TCGA to specifically predict DD-SR interactions between 2,214 genes tested and mTOR as DD rescuer. Each gene pairing (between 2,214 genes and mTOR) was quantified using INCISOR interaction score. The score was used to estimate precision and recall.

Inset in Fig 2A was generated as follows: To obtain normalized counts at each time point, cell counts of each shRNA at each time point were divided by corresponding total number of cell count. To quantify the lethality of vulnerable knockdown in the experiment, we performed a one-sided Wilcoxon rank sum test between initial normalized count with untreated normalized count.

To estimate cell growth rate for each shRNA X, normalized counts were divided by initial normalized count as follows:

$$\text{growth rate}(X) = \frac{\text{normalized count}(X)}{\text{initial normalized count}(X)}$$

Effect of rapamycin treatment on cell growth on knockdown of gene X was calculated as:

$$\text{rapamycin effect}(X) = \frac{\text{treated growth rate}(X)}{\text{untreated growth rate}(X)}$$

### Experimental testing of SR-predicted synergistic drug combinations in head and neck cancer cell lines

Seven drug combinations tested in this experiment were chosen as follows. Among the important cancer genes captured by the DU-SR network, we focused on testing SR interactions between five important HNSC oncogenes (mTOR, PIK3CA, KIT, AKT, and PTK2). INCISOR predicted 5 DU-SR interactions between these oncogenes. Two different inhibitors (rapamycin and INK128) were included for mTOR, and one inhibitor each for PIK3CA, KIT/SRC, AKT, and PTK2 was included in the experiment. This resulted in seven combinations tested (Fig 2B).

Rapamycin was purchased from LC Laboratories (Woburn, MA). Dasatinib, Erlotinib, BYL719, and INK128 were purchased from Selleckchem (Houston, TX). Vita-Blue Cell Viability Reagent was purchased from Biotools (Jupiter, FL). CAL33, HN12, Detroit 562, and SCC47 cell lines were cultured in 96-well plate and then treated with drugs for 48 hours (Raw Data in Dataset Tables EV11–EV17). Assays were performed according to the manufacturer's instructions. Combination index for quantitation of drug synergy was analyzed by CompuSyn software (Chou, 2006, 2010). CI values represent synergism (CI < 1), additivity (CI = 1), and antagonism (CI > 1), respectively (Appendix 4.7).

### Experimental testing of SR-predicted rescuers via siRNA

siRNAs for non-targeting control and PIK3CA were purchased from GE Healthcare (two ON-TARGETplus PIK3CA siRNAs 5′-GCGA AAUUCUCACACUAUU, and 5′-GACCCUAGCCUUAGAUAAA, Lafayette, CO). siRNAs for mTOR were purchased from Sigma (two

MISSION® siRNA human mTOR SASI_Hs02_00338641 and SASI_Hs01_00203144). Cells were cultured in 96-well plate, transfected with Lipofectamine RNAiMAX reagent (Life Technologies, Carlsbad, CA) for 24 h; then, cells were treated with drugs for another 48 h (Raw Data in Dataset Tables EV18 and EV19). Viability assays were completed as previously described (Appendix 4.7). We also validated the knockdown efficiency of PI3K and mTOR siRNA via a Western blot analysis (Appendix Fig S11H) as follows. Cells were transfected with negative control or the corresponding PIK3CA, mTOR (FRAP1) siRNAs. Cells were lysed in lysis buffer (50 mM Tris–HCl pH 7.6, 150 mM NaCl, 1 mM EDTA, 1% Nonidet P-40) supplemented with Halt™ Protease Inhibitor Cocktail (Thermo Fisher Scientific) and phosphatase inhibitors (1 mM $Na_3VO_4$ and 1 mM NaF). Equal amounts of total proteins were subjected to SDS–polyacrylamide gel electrophoresis. Primary antibodies used were from Cell Signaling Technology (Danvers, MA), PI3KCA (catalog number 4255), pAKT (catalog number 2965), AKT (catalog number 9272), pmTOR (catalog number 5536), mTOR (catalog number 2983), pS6 (catalog number 2211), S6 (catalog number 2217), and GAPDH (catalog number 2118), the latter as a protein loading control.

To test whether the KD of INCISOR-predicted rescuers acts as sensitizers, we checked the sensitivity of cells to the primary drug increases upon the KD of the rescuer. The efficacy of the rescuer KD to sensitize cancer cells to a primary drug was estimated as the percentage increase in the sensitivity to the drug following the rescuer KD relative to the sensitivity of the primary drug alone without rescuer KD. Specifically, we used a targeted siRNA to knock down the specific rescuer gene, while an untargeted non-specific siRNA was used as a control. Cell counts were measured for the untargeted/targeted siRNA after primary drug treatments. The normalized response of (targeted/untargeted) siRNA-treated cells to drug treatment was quantified as a change in cell counts relative to the cell counts following the respective siRNA inhibition alone. Next, using this normalized response, DRC was estimated for both targeted and untargeted siRNAs using DRC R-package. Percentage increase in sensitivity of the primary therapy (y-axis, Appendix Fig S8I) due to the rescuer siRNA-KD was estimated as the percentage decrease in IC50 of the combination of primary drug treatment and siRNA inhibition relative to the primary therapy in untargeted siRNA combination. The significance of the increase in drug response was estimated using a standard ANOVA test.

## Drug combination testing of SR interaction involving DNMT1

### Drug combination screen

Drug screening was performed using automated liquid handling in a 1536-well plate format (Friedman *et al*, 2015). The drug doses used were chosen based on previous single agent screening at the Center for Molecular Therapeutics of the Massachusetts General Hospital Center for Cancer Research (Dataset Table EV24).

The screen of two drugs A and B was performed in a $1 \times 5$ format with one dose of drug A combined to five doses of drug B and compared to the effects of the five doses of drug B alone. The five doses of drug B followed a fourfold dilution series (Dataset Table EV24 and EV25).

Cells were seeded at densities optimized for proliferation based on the pre-screen experimental determination in 1536-well plate format. Cells were seeded, placed overnight at 37°C, and drugs added the next day using a pin tool. After 5 days in drug, cells were fixed permeabilized and nuclei stained in a single step by adding a PBS Triton X-100/Formaldehyde/Hoechst-33342 solution directly to the culture medium. Final concentrations: 0.05% Triton X-100/1% Formaldehyde/1 μg/ml Hoechst-33342. Plates were covered and placed at 4°C until imaging.

Imaging was performed on an ImageXpress Micro XL (Molecular Devices) using a 4× objective. Cell nuclei enumeration was performed using the MetaXpress software, and count accuracy was routinely checked visually during acquisition. The screening was conducted in two replicates (two separate 1536-well plates, Dataset Table EV25).

### Calculation of drug combination synergy score

Due to a limited number of dose combination used in the experiments per each drug pair (one concentration of Decitabine (five replicates) and five concentrations of each rescuer inhibitor (two replicates)), Fa-CI analysis is not feasible. The drug dose tested is provided in Dataset Table EV24.

We used Bliss independence model (Bliss, 1939; Lehar *et al*, 2007; Friedman *et al*, 2015) to determine synergistic drug combination which is suitable per such experimental setting. More specifically, to determine Decitabine synergism with a rescuer inhibitor (R) tested in a cell line, we compared following two ratios of experimentally determined cell counts for each dose (C) of rescuer inhibitor (Dataset Table EV25):

$$\text{Ratio(X)} = \frac{Cell\ count(Dectabine + R(C))}{Cell\ count(Decitabine)}$$

$$\text{Ratio(Y)} = \frac{Cell\ count(R(C))}{Cell\ count(Untreated)}$$

where *Cellcount*(X) denotes cell count following the treatment of X. R(C) denotes rescuer inhibitor at dose C. The effect size of synergism at dose C of rescuer inhibitor was estimated as Synergism(R, C) = Ratio(Y)/Ratio(X). This calculation is separately done for each of two replicates of R dose, generating 10 data points (5 dose × 2 replicate) for each rescuer inhibitor.

The final synergism of rescuer inhibitor R was estimated as the median of synergism of the 10 data points, i.e., Synergism(R) = *median*(Ratio(Y) Ratio(X)). Significance of synergism of R was estimated by a Wilcoxon rank sum test comparing Ratio(Y) and Ratio(X) of the 10 data points of R. Finally, R was estimated to be significantly synergistic with Decitabine, if Synergism(R) > 1.25 and *P*-value adjusted for multiple hypothesis corrections is < 0.05 (i.e., FDR < 0.05, Dataset Table EV26).

Analogously, R was considered to be significantly antagonistic with Decitabine, if Synergism(R) < 0.75 and *P*-value adjusted for multiple hypothesis corrections is < 0.05 (i.e., FDR < 0.05, Dataset Table EV26).

## Functional similarity of SR

### Gene ontology similarity

Gene ontology semantic similarity (Yu *et al*, 2010) was used to quantify the similarity of GO terms between a gene pair. When multiple GO terms were associated with a gene, similarity between

all combinations of rescuer GO terms, and vulnerable GO terms were calculated, and the maximum of these scores was taken as final similarity score (average of scores as final similarity gives similar result qualitatively). Distribution of GO similarity of DU-SR pairs was compared with two sets of controls: (i) shuffled network: interactions between rescuer and vulnerable genes of the DU-SR network randomly shuffled, and (ii) random network: gene pairs selected randomly from all protein-coding genes and controlled for similar degree distribution as the original DU-SR network. For each set of control, we determined the similarity measure in an analogous manner as described above for the DU-SR network. Wilcoxon rank sum test was used to calculate the significance of GO similarity of the DU-SR network relative to each control.

### PPI distance

IGraph was used to estimate the distance between two genes in human protein–protein interactions (PPI) network compiled from (Goel *et al*, 2012; Schaefer *et al*, 2012). The PPI distance between gene pairs was compared with two controls, the random network and shuffled network as described above.

### STRING database distance

The STRING network version 10 was downloaded using R-package STRINGdb. STRING database is composed of gene pairs that are likely to share functional similarities. The functional similarity scores provided in the database were estimated using various sources including direct (physical) and indirect (functional) associations. The comparison control networks were made analogous manner as in case of GO similarity described above.

For the DD-SR network, GO similarity, PPI distance, and STRING database distance were estimated analogously (Appendix 3.8).

### PPI-specific DU-SR interactions

To identify DU-SR interactions likely to be mediated by PPI interactions, we applied INCISOR on the human PPI network compiled from (Goel *et al*, 2012; Schaefer *et al*, 2012). The details of the analysis and resultant network are provided in Appendix 3.4. The enrichment of cancer driver genes in the PPI-SR network was calculated using Fisher's exact test.

### Identification of DU and DD rescuers of immune checkpoints

To identify rescuers of immune checkpoints, we removed filtering step 1 (*in vitro* essentiality screens) from INCISOR because the dataset used in step 1 was conducted in *in vitro* models that are deficient of the immune system. Next, we applied the INCISOR to the TCGA to identify the DU and DD rescuers of PD1/PDL1 and CTLA4 downregulation. We call an SR identified by INCISOR as clinically significant if the interaction shows association with survival either in pan-cancer or melanoma cohort in TCGA patient dataset.

### Immunotherapy samples patient samples collection and processing

### Patient samples

A cohort of patients with metastatic melanoma treated was enrolled in clinical trials ongoing at Massachusetts General Hospital for treatment with three immune checkpoint blockades: (i) anti-PD1 or anti-PDL1 (collated together in the analysis and referred as anti-PD1), (ii) anti-CTLA-4, and (iii) combination of anti-PD1 and anti-CTLA-4. Patients were consented for tissue acquisition per Institutional Review Board (IRB)-approved protocol (Kwong *et al*, 2015). These studies were conducted according to the Declaration of Helsinki following informed consent (DF/HCC protocol 11-181) was obtained from all patients.

### RNA sequencing (RNA-seq)

Tumors were biopsied or surgically removed from the consented patients and snap frozen in liquid nitrogen or fixed in formalin. Qiagen AllPrep DNA/RNA Mini or AllPrep DNA/RNA FFPE Kit was used to purify RNA from the frozen or fixed tumor biopsies. RNA libraries were prepared from 250 ng RNA per sample using standard Illumina protocols. Samples were treated with ribo-zero, and then, Epicentre's ScriptSeq Complete Gold Kit was used for library preparation. The quality check was done on the Bioanalyzer using the High Sensitivity DNA Kit, and quantification was carried out using KAPA Quantification Kit. RNA sequencing was performed at Broad Institute (Illumina HiSeq 2000) and The Wistar Institute (Illumina NextSeq 500). BAM files of raw RNA-seq data were used to summarize read counts by featureCounts (Liao *et al*, 2014) with parameters that only paired-ended, not chimeric, and well-mapped (mapping quality $\geq$ 20) reads are counted (Data available online).

Differential expression analysis was conducted by the generalized linear model implementation (McCarthy *et al*, 2012) of R-package "edgeR" and following a standard pre-processing of read count analysis (Zhou *et al*, 2014). Transcript per million (TPM) was used to estimate fold change.

If a patient is treated sequentially with ICBs, A and B and the biopsies are available pre-treatment, post-A-treatment, and post-A + B-treatment. Comparison of pre-treatment vs. post-A-treatment of the patient was considered in the analysis for the resistance of therapy A. In case of multiple biopsies for pre-, on-, or post-treatment are available per patient, all biopsies were considered in the analysis as follows. For the analysis of differential expression using edgeR, an indicator variable per patient was introduced in the design matrix as recommended in the reference manual of edgeR, which controls for individual-specific transcriptome. To calculate the fold change displayed in Fig 6, mean of TPM was taken in case of multiple pre-treatment biopsies per patients; and in case of multiple post- or on-treatment, each biopsy was displayed in the figure (subscripted by ".X[biopsises number]").

## Data availability

We have included the code as Code EV1 (for DU-SR identification). All the data used in INCISOR inference along with code are hosted in homepage at (http://www.umiacs.umd.edu/~vinash85/public/incisor.tar.gz). The more extended version of the code is available on GitHub (https://github.com/vinash85/INCISOR).

**Expanded View** for this article is available online.

### Acknowledgments

We thank Roded Sharan, Max Leiserson, Steve Mount, Noam Auslander, and Justin Malin for their valuable feedback on the manuscript. This research was

supported in part by the Intramural Research Program of the National Institutes of Health, NCI. M.H. is supported by NIH grants 5P01CA114046, 5P50CA174523, and 1U54CA224070, the Dr. Miriam and Sheldon G. Adelson Medical Research Foundation, and the Peer Reviewed Cancer Research Program Grant WX1XWH-16-1-0119 [CA150619].

## Author contributions
ADS, JSL, and ER conceived and designed the research. ADS, JSL, JSG, CB, and ER designed the experimental procedure. ADS and JSL performed the computational analysis and statistical computations. ZW and RI-B performed the experiments. GZ, TT, ZW, BM, NUN, OP, AAF, AA, TM, GK, PG, RKE, LJD, DTF, LJ-A, AW, KC, SGP, WR, KG, GB, SH, MH, CB, JL, and KF collected data used in the paper. All co-authors wrote the paper.

## Conflict of interest
The authors declare no conflict of interests.

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
