## [Review Process File · Molecular Systems Biology]

Genome-wide prediction of synthetic rescue mediators of resistance to targeted and immunotherapy

Avinash Das, Joo Sang Lee, Zhiyong Wang, Gao Zhang, Ramiro Iglesias-Bartolome, Tian Tian, Zhi Wei, Benchun Miao, Nishanth Ulhas Nair, Olga Ponomarova, Adam A. Friedman, Arnaud Amzallag, Tabea Moll, Gyulnara Kasumova, Patricia Greninger, Regina K. Egan, Leah J. Damon, Dennie T. Frederick, Livnat Jerby-Arnon, Allon Wagner, Kuoyuan Cheng, Seung Gu Park, Welles Robinson, Kevin Gardner, Genevieve Boland, Sridhar Hannenhalli, Meenhard Herlyn, Cyril Benes, Keith Flaherty, Ji Luo, J. Silvio Gutkind and Eytan Ruppin.

Review timeline:

Submission date:	18 th March 2018
Editorial Decision:	5 th April 2018
Revision received:	16 th April 2018
Editorial Decision:	22 nd June 2018
Revision received:	14 th October 2018
Editorial Decision:	21 st December 2018
Revision received:	31 st December 2018
Accepted:	18 th January 2019

Editor: Thomas Lemberger

Transaction Report:

1st Editorial Decision

5th April 2018

Thank you again for submitting your work to Molecular Systems Biology and for having now forwarded the related paper that will appear in Nature Communications. I have now had the chance to read both studies and I am afraid we cannot send the MSB submission for review in its present form.

We appreciate of course that the Nat Comm study addresses the question of syntehtic lethal pairs whereas the MSB study explores the concept of synthetic 'rescuing' interactions and their potential impact on resistance. However the methodolgy used in both studies is essentially the same: 1) starting from candidate pairs identified in vitro (16 Mio SL candidates, much less? SR candidates); 2) expression and CNV data from TCGA to identify under-, resp. over-represented pairs; 3) association with better, resp. worse, patient survival; 4) phylogenetic similarty.

Given the similarity of theses steps (pending the sign of the filtering at step 2 and 3), we are not convinced that the presentation of INCISOR as "a new data mining approach" (page 4) is supported.

We appreciate that the follow up analyses are however different.

We would thus invite you to rewrite part of the study to put the method it its proper context and clarify the aspect of novelty:

- the general strategy of the data mining method should be first explained in the Introduction and clearly associated with the Lee et al study.

- Results should not present the data mining strategy as novel but explain in what sense it has been modified as compared to the SL approach (overrepresentation instead of underrepresentation in step 2, worse rather than better survival) to make clearer what is the precise contribution of the submitted study.

- A better discussion of the SL vs SR network could be useful in Discussion, given that the SL network seems to be much larger than the SR set, and to provide a comparison and discussion of the accuracy of the method in both SL and SR scenarios.

Please resubmit your revised manuscript online, with a covering letter listing amendments and responses to each point raised by the referees.

1st Revision - authors' response

16th April 2018

Thanks for considering our manuscript (Genome-wide prediction of synthetic rescue mediators of resistance to targeted and immunotherapy, MSB-18-8323) for *Molecular Systems Biology*. In accordance with your suggestions, we have made following changes to the manuscript:

In the introduction, we describe a strategy to mine patient genomic and survival data to identify SL interactions via ISLE (Lee et al.). We then present INCISOR as a modified version of ISLE that is specifically designed to identify SR interactions, as requested.

In the Result sections we now further recount that INCISOR is not a novel data mining approach on its own but a modification of ISLE. We have explicated the differences between ISLE and INCISOR in the Methods section. Further, we have added an analysis comparing the performance of INCISOR to that of ISLE in predicting patients drug response across the TCGA, pointing to the superiority of INCISOR in that task (Figure 6).

Finally, in the Discussion section, we now explicitly discuss the difference in the goals of ISLE vs INCISOR in predicting response vs. emerging adaptive resistance to cancer therapy. Interestingly, we also now demonstrate that SL and SR interactions can be combined to further improve the survival prediction of breast cancer patients (the comprehensive study and testing of the power of such combinations is of course a whole different future study on its own) .

2nd Editorial Decision

22nd June 2018

Thank you again for submitting your work to *Molecular Systems Biology*. We have now heard back from the two referees who agreed to evaluate your manuscript. As you will see from the reports below, the referees find the topic of your study of potential interest. They raise, however, substantial concerns on your work, which, I am afraid to say, preclude its publication in its present form.

The major points raised by the reviewers refer to the following issues:

- tumor-specific confounders may possibly represent an important flaw in the analysis; this issue should be particularly carefully and convincingly addresses as it may seriously alter the conclusions of the analysis.
- multiple testing issues
- demonstration of the robustness of the analysis to the thresholds chosen and motivation of those thresholds.
- in the experimental part, reviewer #2 also raise important concerns with regard the the lack of specificity of the inhibitors used and the need for more rigorous validation.

REFeree REPORTS

Reviewer #1:

The manuscript describes comprehensive discovery of genetic interactions in cancer with a focus on synthetic rescue interactions. The authors integrate a multitude of omics datasets from patients and

cell lines and perform a series of experimental validations of their computational analyses, including responses to chemo- and immunotherapies, in what appears to be an impressive piece of systems biology research overall.

I have one major concern about the computational analysis that may indicate a large flaw affecting their results and conclusions throughout the paper. Specifically, most or all analyses are conducted in a pan-cancer fashion such that molecular heterogeneity of cancer types is not accounted for; this likely causes systematic biases.

For instance in the survival-of-the-fittest analysis, many gene pairs where up-regulation of one gene is accompanied by down-regulation of another gene may occur because the activities of specific pathways and gene regulatory programs differ in types of cancer. Further, the gene activity cutoffs seem to be arbitrary (e.g. 33rd percentile).

Some potential remedies: in statistical analyses, using a regression model with cofactors for cancer types that would soak up the effect of cancer types for testing the effect of the genetic interaction partner, and constructing one or few additional genetic interaction networks for specific cancer types (for example, a couple of largest cohorts with patient and cell line data) to demonstrate the robustness of their approach in recapitulating the pan-cancer results.

This comment applies to other areas of the study as well. For example, survival analysis should be conducted for every cancer type separately.

Reviewer #2:

Das et al. introduce the INCISOR pipeline, which combines a diverse set of large-scale data analyses to generate a final list of putative SR (Synthetic Rescue) interaction pairs. "Synthetic rescue" is defined as those genes that, when over- or under-expressed, promote the survival of a "vulnerable" gene in a cancer cell. For example, the authors argue that decreased activity of gene X can be buffered by increased expression of genes Y, Z, etc., a relationship that they term DU (vulnerable gene Down, rescue gene Up)-SR, or by decreased expression of genes A, B, C etc., which they designate as DD (vulnerable gene Down, rescue gene Down)-SR, respectively. UD-SR and UU-SR relationships follow a similar logic. To define these interactions, they incorporate disparate and often separately mined data types, including shRNA knockdown screens and drug response data, co-occurrence of aberrant genomic events, multivariate patient survival analysis and measures of phylogenetic similarity. These analyses are combined sequentially, acting as successive filters to whittle down an initial set of SR candidates obtained from an analysis of published essentiality and drug response screens. The authors then present a diverse range of analyses to make the case for the utility of the INCISOR SR predictions. These include experiments examining potentially synergistic drug combinations in cancer cell lines, large-scale data mining of TCGA patient drug response and survival data, analyses of drug response data from the CCLE and other data sources, comparison to previously published methods, the analysis of shared SR functional annotations and the proximity of SR interaction pairs in published PPI networks.

This paper develops an interesting concept, and as a whole, the INCISOR pipeline could provide a valuable tool for discovery/hypothesis generation and testing. INCISOR might also provide a valuable tool to mine recently published, large-scale essentiality datasets such as the Novartis DRIVE shRNA compendium and the Broad Institute's Avana CRISPR screens. In short, the INCISOR approach will probably be a useful addition to currently available pipelines used to extract testable hypotheses from the large but unwieldy corpus of publicly available genomics data. However, the current version of the manuscript and the analysis could be substantially improved, both conceptually and technically. In particular, many of the kinds of analyses usually presented for similar types of algorithms (e.g., robustness/sensitivity type assays) are either not presented or presented and not really discussed. Attention should be given to the following specific issues in a revised manuscript.

Major Points:

1) The sequential ordering of the INCISOR steps suggests that the largest number of SR candidates are produced by step 1, and further filtered by subsequent analyses. The authors provide a rationale for this process, but they do not discuss in the paper how much each contributes to the power of INCISOR. They do show how the various steps contribute in Supplementary figure 3, but they do not really describe this figure in any detail, nor do they provide any explanation of why different parts of the pipeline have significantly different contributions to DU vs DD vs UD vs UU SR interactions, respectively. We need to see some sort of multivariate analysis where they use various combinations of the four steps in the pipeline and convince us that one needs (or benefits from) all four to get maximal predictive power.

2) Related to the above, could the authors give ball-park figure for the number of candidates produced/filtered at each step? If my understanding of the pipeline description is correct, roughly $20,000^2$ sets of genome-scale (10-15000) Wilcoxon tests are performed. This is an impressively thorough canvassing of possible co-aberrations, but also raises a massive multiple-testing issue, due primarily to the 20,000² co-aberrations. Obviously, in a hypothesis generation/discovery context, the emphasis should be on hypothesis generation over a narrow adherence to statistical significance. That said, can the authors elaborate on the way in which they applied multiple-testing adjustment to obtain their FDR-adjusted p-values?

3) Also related to point 1, if in fact, all steps of the pipeline contribute substantially to the predicted power of INCISOR, then how can the modification used to find potential immunotherapy SRs, which basically omits step 1 and keeps steps 2-4 unchanged, possibly have anywhere near the predicted power of the full pipeline used for predicting small molecule and gene SRs? And for I/O, several of the targets that they identify (e.g., CTLA4, KIRs) are in infiltrating immune cells, NOT in the tumor cells themselves. How, then, does the tumor purity correction used for TCGA data analysis affect this assessment?

4) In step 1 of the INCISOR pipeline, the authors describe the use of a univariate (Wilcoxon) test to identify differential essentiality or differential drug response in the context of pairwise genomic aberrations. In both shRNA and drug response screens, tumor type-specific essentiality/drug response is a primary confounding factor when one considers genomics-associated essentiality/response. For example, Marcotte et al., 2016 report thousands of differentially essential gene predictions in pairwise comparisons of different cancer types, while reporting dozens or hundreds of genes whose essentiality is associated with expression or copy number differences. Have the authors considered using a multi-variate analysis approach that would allow incorporation of this important potential confounder when testing for differences associated with paired genomic aberrations? Cancer type confounding seems to have been correctly identified and incorporated as an explanatory variable in the multivariate Cox regression used in Step 3, but also seems largely missing from Step 2 of the INCISOR pipeline.

5) In Step 2 of the INCISOR pipeline (co-occurring genomic aberrations), using tertiles to separate expression values into over/under-active classes seems a quite a coarse classification. Given the high-throughput nature of this analysis pipeline, thresholding decisions are understandable. Furthermore, this tertile classification approach is also used in the published PARADIGM algorithm for classifying activity levels of genes. That said, have the authors considered alternative approaches, such as using (potentially simple) expression outlier-detection methods, to classify over/under activity? The tertile approach seems poorly suited to the large number of genes that show relatively little variation across samples, or that are either not expressed or expressed in a small fraction of samples. It seems that these thousands of genes would also be examined for potential "co-aberrations", further inflating the large multiple-testing challenge posed by the sheer number of comparisons performed by the pipeline. In any event, it would seem incumbent on the authors to provide a sensitivity/robustness analysis to show how their results are affected if they take top 25%/20% etc. of genes instead of top tertile.

6) The authors use of SCNA and gene expression data for different steps in the INCISOR pipeline seems inconsistent-or at least it is confusing. On p. 23, they state that "We divide cell lines into those having high or low expression of gene R... and then "pairs that pass both (functional) tests either using gene expression OR SCNA are referred to as putative SR. So here, they take the UNION of gene expression and SCNA. But then in using the TCGA data on p. 24, they mandate that a gene be both over-expressed and have an SCNA (i.e., the intersection of these two sets). Why do they use both one place and either in the other?

7) The different SR networks differ significantly in sparsity-at least by eye. Can the authors explain why this might be?

8) The authors state that they used 2200 genes in their shRNA in the H/N cancer line HN12. What was the rationale for the choice of 2200 as opposed to entire transcriptome? What are the properties of the genes in this library? Also, they state that they infected at an MOI of 1. Why? This is too high-at MOI of 1, a significant number of cells should receive multiple hairpins. Typically, such screens are done at MOI=0.3 or lower. In addition, inadequate information is given for us to evaluate the validity of this screen-especially the in vivo component. It is well known that only a fraction of the cells that are xenografted actually engraft, which creates a natural "bottleneck" on the shRNA pool. If they just inject cells infected with a neutral barcode library, how many barcodes do they recover (i.e., without any selection)? Put another way, what is the tumor-initiating frequency of this line? Without such information, it is impossible for us to assess this in vivo screen.

9) Also related to the screen, in Figure 2a, the authors show the precision and recall at the threshold where INCISOR identifies 75 genes-why did they pick this number? As written, the reason for this choice is obscure (at least to me).

10) In Figure S2g and h, the authors purport to show the functionally active SR genes "as cancers progress" but it is not clear how they are defining progression-and what are the units on the X axis here? In Figure S2i, they argue that tumors with a high number of SRs, which they define as top 10%, do better than those with lowest 10%--again, some sort of robustness analysis is needed-what happens if they pick highest/lowest 5%, 15%, 20%?. Also, in Figure S2j, they compare survival combining SR and SL, but they don't show us survival with just SL-why not?

11) It is not clear what the authors are trying to convey with Figure S2L-this figure shows substantial differences in the number of SRs in different cancers-but what is their argument-surely SRs are not the only determinant of response, as for many of these cancers, whether or not they are drug sensitive is dispositive. For example, ovarian cancer might have one behavior unperturbed-which could depend on SRs for the genetic changes that cause the tumor-but whether or not a given ovarian cancer patient survives or dies is largely dependent on whether they have HR deficiency (and thus are cis-PLT sensitive) or not. Similarly, whether a breast cancer patient survives or not can depend on HER2 and ER status.

12) On p. 8, the authors test a number of drug combinations predicted by their SR networks, and claim that several are synergistic. However, two of the combinations involve dasatinib, which inhibits MULTIPLE kinases, not just KIT and SRC as they imply. How can such a broad spectrum kinase inhibitor be used in ny way to infer specificity. Similarly, their PTK2 inhibitor also inhibits PTK1-how can they infer specificity. Similar problems attend the use of other very broad spectrum kinase inhibitors, such as Sunitinib and Sorafenib, which are most certainly NOT (only) VEGFR and RAF inhibitors, respectively.

13) Related to point 10, the authors state that rapamycin is a TOR inhibitor, but it is only a TORC1 inhibitor-Torin would be a better choice to test if one is claiming to represent TOR inhibition.

14) The authors state that "to predict the response of an individual patient's (sic) to a given drug, we defined the drug-tumor SR score as the number of upregulated rescuers of a drug's target in that patient's tumor.." OK-but why the number? Wouldn't the strength of the SR be relevant-i.e., are two strong SRs better than 100 weak ones? Why should it just be the number-or better still, why don't the authors provide some type of analysis justifying this choice?

15) In Figures S4f, g, what does "normalized IC50" mean-how is this normalized? Also, I don't understand the data presentation in Figure S4h. The legend says, "The phenotypes measured are the post-knockdown growth rates in pooled shRNA screening." What do they mean? What one measures in pooled screens is enrichment for, or depletion of, barcodes representing different shRNAs-one doesn't measure growth rate. Did they test some number of hits from the screen and measure growth rate? Please explain.

16) The validations in the TOR experiments (and others) are sub-optimal. We should at least see a few cases where they did knockdown and rescue (the only compelling proof) or an orthogonal type of experiment like CRSPR/Cas9 deletion or dominant negative. At the very least, they should show knockdown efficiency so we can see that the biological effect parallels the extent knockdown.

17) What are the authors trying to convey in Figure S10h/I when none of the differences appear to be statistically significant?

Minor comments:

- 1) There are inconsistencies in nomenclature at various places in the paper: e.g., sometimes the authors use DD-SR, other times SR-DD. Please be consistent.
- 2) Similarly, in some of the figure headings, all words are capitalized; in others, just the first one is.
- 3) The authors start several sentences with numbers-please type out the number if it begins a sentence.
- 4) On p. 4, the authors describe DU-SR genes first and then DD-SRs, yet in the figure, they show DD in middle and DU on right-I would reverse this order
- 5) It is difficult to read the X and Y axes on many of the figures; it is also difficult to read some of the figures (especially the interaction diagrams). Please print out every figure and make certain that each can be read easily by someone with 20:20 vision (not X-ray vision!).
- 6) Also, if one clicks on a node in a network diagram, one should be able to see the identify of the gene-only some of the genes are labelled in their network diagrams, and the rest are seen only as spots. Please correct this.
- 7) On p. 8, they authors state that mTOR can activate AKT independently of PI3K, and cite a review as a reference-as this is certainly NOT generally believed to be the case, please cite the primary reference.
- 8) On p. 10, the authors state that DNMT1 is a "key oncogene in non-small cell lung cancers." DNMT1 is over-expressed in many such tumors, but I am unaware of any evidence showing that it

meets the criteria for an oncogene-please provide a reference to support this claim. Also, the pathways for DNMT1/b-catenin, etc., are not all well validated and should probably be presented as plausible or possible pathways to explain the data, not gospel.

9) Figure 3 D/E - Given the discrete nature of these multinomial distributions, each consisting of a small number of possible integer values (1-6), a boxplot representation useful for continuous (or many-valued discrete) distributions seems inappropriate. A more suitable visualization may be side-by-side bar charts. Also, "protein" is mis-spelled in Figure 3D.

10) The legend to Figure S2n makes no sense as written. Please correct

11) For Figure S3, please label the drugs on the figure-what is "BRAF inhibitor?" Also, what is "estrogen receptor inhibitor." Do they see differences between Vemurafenib and "BRAF inhibitor" in V600E melanoma lines-and if so, can they explain why? Ditto for tamoxifen and estrogen receptor inhibitor in ER+ breast cancer (i.e., can they find SRs that are different for these two drugs)?

12) The authors forgot to plot the "Random" line on Figure S4B.

Reviewer #1:

The manuscript describes comprehensive discovery of genetic interactions in cancer with a focus on synthetic rescue interactions. The authors integrate a multitude of omics datasets from patients and cell lines and perform a series of experimental validations of their computational analyses, including responses to chemo- and immunotherapies, in what appears to be an impressive piece of systems biology research overall.

Thanks.

I have one major concern about the computational analysis that may indicate a large flaw affecting their results and conclusions throughout the paper. Specifically, most or all analyses are conducted in a pan-cancer fashion such that molecular heterogeneity of cancer types is not accounted for; this likely causes systematic biases.

For instance, in the survival-of-the-fittest analysis, many gene pairs where up-regulation of one gene is accompanied by down-regulation of another gene may occur because the activities of specific pathways and gene regulatory programs differ in types of cancer.

Thank you for this important comment. In the previous version, we already controlled for cancer type expression in the survival of fittest (step 2) of INCISOR as follows: We normalized the expression within each cancer type, i.e., within each cancer type, an equal fraction of samples (one third) were assigned to be up-regulated (or down-regulated). We now additionally show that using a mixed linear model to control for cancer type does not change our SR predictions. Further, cancer type specific survival differences were also already implemented in the previous version of INCISOR. However, following your comment, we decided to deepen this further and now explicitly control for different cancer types by introducing a linear mixed model in step1 (see below). Overall, we found that 162 (DU) SR interactions identified previously were potentially confounded and have removed them from the analysis. We have now re-conducted all analyses reported in the manuscript using the new set of 1033 SR-DU interactions (and filtered the DD interactions in a similar manner, resulting in the removal of 1967 interactions).

Further, the gene activity cutoffs seem to be arbitrary (e.g. 33rd percentile).

We chose the 33rd percentile because of simplicity and convenience of comparing p-values across different gene pairs. It also seems a natural choice since we consider three levels of activity (down, WT and up) for each gene, as usually is conceived by biologists. Following your comment, we have now tested the robustness of this cut-off by rerunning INCISOR using a 25% threshold for gene activation (i.e. top 25% of samples were considered over-expressed, while the bottom 25% were considered down-regulated and the rest 50% as WT). The result is shown in figure R1. The overlap of the SR pairs identified using these thresholds is highly marked and significant (Hypergeometric $P < 2.2E-16$). This analysis has been now included in Appendix section 2.4.

Figure R1: Overlap of predicted (DU) SR pairs using INCISOR with gene activation thresholds of 0.33 and 0.25.

Some potential remedies: in statistical analyses, using a regression model with cofactors for cancer types that would soak up the effect of cancer types for testing the effect of the genetic interaction partner,

Thanks. We have now followed-up on your suggestion and control for cancer type by using a linear mixed model. These results are now briefly referred from the main text on page 5 and described in more detail in method section on page 20. We now control for cancer types as follows:

For input datasets in which in vitro experiments were conducted across multiple cell lines, we model cancer types of cell lines as a random effect in the mixed linear model:

$$y \sim g + (1|cancer_type)$$

Where y is the effect of knockdown of a vulnerable gene in cell proliferation, and g is the expression of rescuer gene. $(1|cancer_type)$ represents random effect of the confounding factor. y is quantile normalized to $N(0,1)$. The P-values of the coefficient of rescuer gene expression/SCNA, was estimated using ANOVA and were adjusted for multiple hypotheses by calculating the false discovery rate on number of hypotheses tested in each dataset. Pairs significant in any dataset (either using gene expression or SCNA) were selected as putative SR pairs. The mixed model resulted in 1033 DU-SR pairs and 1967 DD-SR pairs. We have updated the results of the manuscript using this new set of more robust predicted SRs in all analyses reported.

In addition to normalizing the gene expression within each cancer type, we also now explicitly control for cancer type in step 2 (molecular screen) using a mixed model analogous to the model described above (i.e., we modeled expression of a vulnerable gene as the mixed effect of its rescuer gene and cancer type). However, all the identified pairs remain significant following the new modeling in step 2 and do not affect final SR set predicted.

... and constructing one or few additional genetic interaction networks for specific cancer types (for example, a couple of largest cohorts with patient and cell line data) to demonstrate the robustness of their approach in recapitulating the pan-cancer results

Thanks, following up on your suggestion, we have now applied INCISOR to TCGA 1098 breast cancer (BC) patient data to identify the DU-SR networks specific to breast cancer. We have chosen breast cancer as it has the largest cohort in the TCGA collection, and has another large independent

cohort METABRIC(*I*) on which we could test the emerging predictions in an independent manner. In vitro filtering (step 1) of INCISOR was conducted using subset of in vitro screens performed on breast cancer cell lines. The details of the analysis are provided in Appendix section 3.11. INCISOR identified 419 DU interactions in breast cancer (Dataset Table S32), 123 of these interactions overlap with pancancer DU-SR interactions. The breast cancer rescuers are enriched in EGFR tyrosine kinase inhibitor resistance, PI3K-AKT, RAS, Jak-stat and MapK signaling (Dataset Table S34). The vulnerable genes are enriched in adherens junction, proteoglycan, and carbon metabolism (Dataset Table S34). The interactions were predictive of patient survival, which we validated using Metaberic dataset (Fig R2a). We also studied the ability of the breast cancer DU-SR network to predict the clinical response of 3873 patients in the TCGA dataset for which we have drug-response data, using the exact same procedure reported in the main text (there using the pan cancer SRs). As demonstrated, breast cancer DU-SR accurately classify patients into responder and non-responders for 15 of the drugs (Fig R2c, vs 22 drugs predicted via the pan-cancer inferred SRs, as reported in the main text).

We also applied INCISOR to identify DD-SR interaction specific to breast cancer. We identified 341 DD-SR interactions, 89 of these interactions were common with pancancer DD-SR interactions. The complete list of DD breast cancer specific interactions is provided in Dataset Table S33. The breast cancer DD rescuers and vulnerable genes are enriched in PI3K-AKT signaling, human papillomavirus infection and cell cycle (Dataset Table S35). The breast cancer DD-SR interactions also predict patient survival in Metaberic dataset (Fig R2b).

We have now included the details in Appendix section 3.11, and briefly refer to those in the main text (page 7) as follows:

“...we applied INCISOR to breast cancer in vitro screens and breast invasive carcinoma (BRCA) patient data from TCGA to identify the breast cancer specific DU and DD SR interactions. The resultant breast cancer SR network is shown to be predictive of breast cancer patients survival and to a lesser extent, to be predictive of patients drug response across different cancer types.”

Fig R2: **Breast cancer SR interaction:** (a,b) Patients survival prediction: Breast cancer DU-SR (a) and DD-SR (b) network predict patient's survival in the Metaberic dataset. A Kaplan-Meier (KM) analysis comparing the survival of patients whose tumors have many functionally active SRs (top 10 percentile (N=200), rescued) to those with a few (bottom ten percentile (N=200), non-rescued). The difference in the areas under the curve between rescued (blue) and non-rescued (red) samples (Δ AUC) and their log rank p-values are denoted, in addition to Hazard ratios and their significance obtained from a Cox regression. (c) The Y axis displays Logrank p-values per drug, denoting how well response is predicted by DU-SR network in terms of survival difference between predicted responder and non-responders.

This comment applies to other areas of the study as well. For example, survival analysis should be conducted for every cancer type separately.

Thanks. The robust inference of SRs requires a large number of samples that yet does not exist for many cancer types and hence we strived first and foremost to identify a pan cancer network composed of interactions that are common across different cancer types. However, as suggested by the reviewers and addressed by us, INCISOR carefully controls for the potential confounding effects of different cancer types when inferring the pan cancer network. Specifically, the survival analysis step in INCISOR is controlled for cancer type in the previous version by including cancer type as co-factor (patient stratification) in Cox regression. This stratified cox-regression allows likelihoods to be calculated *for each patient cancer-type separately* and only the final likelihood of the model is average of likelihood across all cancer-types. Specifically, we model patient survival as:

$$h_g(t, \text{patient}) \sim h_{0g}(t) \exp(\beta_1 I(V, R) + \beta_2 g(V) + \beta_3 g(R) + \beta_4 \text{age} + \beta_5 \text{GII} + \beta_6 \text{TP})$$

Where g is a variable over all possible combinations of patients' stratifications based on cancer-type, race, and sex. h_g is the hazard function. Other confounding factors like sex, race, etc are controlled in a similar fashion. We have clarified this now further in the revised main text (page 22) accordingly as follows:

“The above modeling of survival as stratified cox regression allows to account for systematic differences in survival in different cancer types. INCISOR assumes a different baseline hazard for each cancer type to compute likelihoods. The estimated likelihoods are then combined to estimate the effects of gene interactions on survival.”

Reviewer #2:

Das et al. introduce the INCISOR pipeline, which combines a diverse set of large-scale data analyses to generate a final list of putative SR (Synthetic Rescue) interaction pairs. "Synthetic rescue" is defined as those genes that, when over- or under-expressed, promote the survival of a "vulnerable" gene in a cancer cell. For example, the authors argue that decreased activity of gene X can be buffered by increased expression of genes Y, Z, etc., a relationship that they term DU (vulnerable gene Down, rescue gene Up)-SR, or by decreased expression of genes A, B, C etc., which they designate as DD (vulnerable gene Down, rescue gene Down)-SR, respectively. UD-SR and UU-SR relationships follow a similar logic. To define these interactions, they incorporate disparate and often separately mined data types, including shRNA knockdown screens and drug response data, co-occurrence of

aberrant genomic events, multivariate patient survival analysis and measures of phylogenetic similarity. These analyses are combined sequentially, acting as successive filters to whittle down an initial set of SR candidates obtained from an analysis of published essentiality and drug response screens. The authors then present a diverse range of analyses to make the case for the utility of the INCISOR SR predictions. These include experiments examining potentially synergistic drug combinations in cancer cell lines, large-scale data mining of TCGA patient drug response and survival data, analyses of drug response data from the CCLE and other data sources, comparison to previously published methods, the analysis of shared SR functional annotations and the proximity of SR interaction pairs in published PPI networks.

This paper develops an interesting concept, and as a whole, the INCISOR pipeline could provide a valuable tool for discovery/hypothesis generation and testing. INCISOR might also provide a valuable tool to mine recently published, large-scale essentiality datasets such as the Novartis DRIVE shRNA compendium and the Broad Institute's Avana CRISPR screens. In short, the INCISOR approach will probably be a useful addition to currently available pipelines used to extract testable hypotheses from the large but unwieldy corpus of publicly available genomics data. However, the current version of the manuscript and the analysis could be substantially improved, both conceptually and technically. In particular, many of the kinds of analyses usually presented for similar types of algorithms (e.g., robustness/sensitivity type assays) are either not presented or presented and not really discussed. Attention should be given to the following specific issues in a revised manuscript.

Major Points:

1) The sequential ordering of the INCISOR steps suggests that the largest number of SR candidates are produced by step 1, and further filtered by subsequent analyses. The authors provide a rationale for this process, but they do not discuss in the paper how much each contributes to the power of INCISOR. They do show how the various steps contribute in Appendix figure 3, but they do not really describe this figure in any detail, nor do they provide any explanation of why different parts of the pipeline have significantly different contributions to DU vs DD vs UD vs UU SR interactions, respectively. We need to see some sort of multivariate analysis where they use various combinations of the four steps in the pipeline and convince us that one needs (or benefits from) all four to get maximal predictive power.

Thank you for the important comment. To address it, we have now included new tables in the paper that show the contribution of each step to INCISOR (see Table R1 below). We also have included the multivariate analysis suggested: To assess the contribution of each screening step of INCISOR, we build the following multivariate logistic regression to predict whether a gene-pair is SR or not. The

model uses the (log) of p-values from the four INCISOR steps to predict known SR interactions corresponding to four drugs compiled from literature survey (described in the main text on page 7). We then evaluate the prediction power of each step to the overall prediction. Fig R3b. summarizes the results of this multivariate analysis. The latter corroborates the univariate analysis done earlier (Fig R3a). Because the contribution to the overall prediction of each screen is different across datasets, combining these screens is a good strategy. As these figures demonstrate, all four screens contribute to the final prediction in INCISOR. Note that the analysis also suggests the possibility that, with the future availability of larger training datasets, a supervised model could further improve the performance of INCISOR. The revised text in discussion (page 17) reads: “Using a multivariate logistic regression we also show that each screening step of INCISOR contributes to its overall predictive power. These contributions vary across different datasets, testifying that combining the screens is a good strategy. This analysis also suggests that, with the growing availability of additional validated SR interactions for training, the performance of INCISOR could be possibly be improved in the future by adopting a supervised strategy.”

Figure R3: The contribution of each step of INCISOR to its overall prediction, assessed via univariate analysis for four published datasets identifying DU-SR rescuers of four drugs (2-6). The AUCs reported quantify INCISOR accuracy in identifying the rescuers of each of four drugs screened,

including ABT-737, BET inhibitor, Lapatinib and an Estrogen receptor inhibitor. (b) The coefficients estimated (Y-axis) by a multi-variate logistic regression for each INCISOR step, for the same four datasets. The significance of the coefficients is displayed (“*” < 0.05, “**” < 0.01, “***” < .001).

2) Related to the above, could the authors give ball-park figure for the number of candidates produced/filtered at each step? If my understanding of the pipeline description is correct, roughly $20,000^2$ sets of genome-scale (10-15000) Wilcoxon tests are performed. This is an impressively thorough canvassing of possible co-aberrations, but also raises a massive multiple-testing issue, due primarily to the 20,000² co-aberrations. Obviously, in a hypothesis generation/discovery context, the emphasis should be on hypothesis generation over a narrow adherence to statistical significance. That said, can the authors elaborate on the way in which they applied multiple-testing adjustment to obtain their FDR-adjusted p-values?

Thanks for this important comment. The first step of INCISOR uses both shRNA and drug response screens to compile candidate in vitro SRs. As pointed out, we are testing 15486 x 19001 (shRNA KD) hypotheses and we have adjusted the p-value using Benjamini-Hochberg and apply FDR < 0.2 accordingly, to identify 2,878,319 putative (DU) shRNA screens derived interactions. In the case of drug response, the hypothesis space is smaller, with 221(number of drugs) x 19001 KDs. We again applied a uniform FDR < 0.2 threshold and identified 354K putative drug SR interactions. We have now included a summary table of these results. The first step ends up identifying 3 million (DU) SR candidates; the second step identified 1.2 million pairs, the third step reduced selected 9021 pairs, followed by 1033 interactions in the final step. These results are now described in Table R1 below, which we have added to the main text (page 23) and Dataset Table S31.

	DU	DD
Steps	Number of pairs	Number of pairs
Total gene pairs analyzed	19001 * 19001 = 361038001 (Note SR is an asymmetrical interaction)	19001 * 19001 = 361038001 (Note SR is an asymmetrical interaction)
Step 1 (In-vitro screen)	3029809	23610190
Step 2 (Molecular screen)	1192838	813669
Step 3 (Clinical screen)	9021	7560
Step 4 (Phylogenetic screen)	1033	1967

Table R1: Number of pairs produced following each step in INCISOR for DU-SR and DD-SR interactions.

Multiple hypothesis corrections were independently applied for each step. To combine p-values obtained via analyzing the multiple datasets of shRNA and drug screens that were used step 1, we tested two alternatives: (1) We first calculate the adjusted p-value within each dataset, and then we applied a multiple hypothesis correction on the adjusted p-value for each pair tested. (2) We also studied the Fisher method to rigorously combine p-values across multiple datasets in step 1 and then applied FDR on combined p-values. The results from both alternatives approach were identical. These details have now been included in the methods section on page 23.

3) Also related to point 1, if in fact, all steps of the pipeline contribute substantially to the predicted power of INCISOR, then how can the modification used to find potential immunotherapy SRs, which basically omits step 1 and keeps steps 2-4 unchanged, possibly have anywhere near the predicted power of the full pipeline used for predicting small molecule and gene SRs? And for I/O, several of the targets that they identify (e.g., CTLA4, KIRs) are in infiltrating immune cells, NOT in the tumor cells themselves. How, then, does the tumor purity correction used for TCGA data analysis affect this assessment?

Thanks. We agree that following the exclusion of step 1 in identifying SRs related to immunotherapy (ICB) may indeed lead to a higher false positive rate in immunotherapy SRs predictions as compared to targeted therapy SR predictions. Due to lack of appropriate data, the effect of the higher false positive rate could not be estimated. Accordingly, we now acknowledge that (page 17): “We expect a higher false positive rate from INCISOR when predicting SRs of ICB treatments because the first screening step cannot be performed.” As you rightfully pointed out, the bulk RNAseq cannot reveal whether the observed expression changes occurred in tumor or immune cells. As Immune infiltration is likely to affect expression of multiple genes in bulk tumor RNA-Seq and may be associated with patient survival interpedently, we have controlled for tumor purity in INCISOR. In the future, analyses of single cell RNA-Seq (scRNAseq) could further increase INCISOR’s accuracy but such data is not yet available on a large scale.

4) In step 1 of the INCISOR pipeline, the authors describe the use of a univariate (Wilcoxon) test to identify differential essentiality or differential drug response in the context of pairwise genomic aberrations. In both shRNA and drug response screens, tumor type-specific essentiality/drug response is a primary confounding factor when one considers genomics-associated essentiality/response. For example, Marcotte et al., 2016 report thousands of differentially essential gene predictions in pairwise

comparisons of different cancer types, while reporting dozens or hundreds of genes whose essentiality is associated with expression or copy number differences. Have the authors considered using a multivariate analysis approach that would allow incorporation of this important potential confounder when testing for differences associated with paired genomic aberrations? Cancer type confounding seems to have been correctly identified and incorporated as an explanatory variable in the multivariate Cox regression used in Step 3, but also seems largely missing from Step 2 of the INCISOR pipeline.

Thanks. We agree and we now explicitly control for cancer types in step 1 (considering shRNA and drug screens) too, as suggested by the reviewer. This was done using a linear mixed model as follows: For input datasets in which *in vitro* experiments were conducted across multiple cell lines, we model cancer types of cell lines as a random effect in the mixed linear model:

$$y \sim g + (1|cancer_type)$$

Where y is the effect of knockdown of a vulnerable gene in cell proliferation, and g is the expression of rescuer gene. $(1|cancer_type)$ represents the random effects of the confounding factor. y is quantile normalized to $N(0,1)$. P-value of the coefficient of rescuer gene expression/SCNA, was estimated using ANOVA. Pairs significant in any dataset (either using gene expression or SCNA) were considered for further analyses after correcting for multiple hypotheses tested across the dataset. The mixed model resulted in 1033 DU-SR pairs and 1967 DD-SR pairs. We have updated all the results of the manuscript based on the new set of SRs (which are a more strict subset of the SR set used in the previous version). The technical details of linear mixed model are added to method section and briefly described in page 5 of main text. They read as follows :

“To determine this association between V and R while controlling for cancer-types of cell lines used in the screens, INCISOR uses a linear mixed-effects (46) model. P-values of association were determined using ANOVA and corrected for multiple hypotheses tested. For each input screen, we model cancer types of cell lines as a random effect in the linear mixed-effects model (46). Specifically, we model the effect of a vulnerable gene knockdown (y) on cell proliferation as a linear mixed-effects model of its rescuer expression (g) and cancer type, where g is modeled as fixed effect and cancer type is modelled as random effect as follows:

$$y \sim g + (1|cancer_type)$$

Here we follow the standard notation $(1|cancer_type)$ to represent the random effect of the confounding cancer type. In the case of shRNA screens y represents gene essentiality of the

vulnerable and in the case of drug screens y represents the IC50 of a drug that inhibits the vulnerable gene. y is quantile normalized to $N(0,1)$ and parameters are estimated using the lme4 software (46). The P-value of the fixed effect was estimated using ANOVA. P-values were adjusted for multiple hypotheses by calculating the false discovery rate considering the number of hypotheses (pairs) tested in each screen.”

5) In Step 2 of the INCISOR pipeline (co-occurring genomic aberrations), using tertiles to separate expression values into over/under-active classes seems a quite a coarse classification. Given the high-throughput nature of this analysis pipeline, thresholding decisions are understandable. Furthermore, this tertile classification approach is also used in the published PARADIGM algorithm for classifying activity levels of genes. That said, have the authors considered alternative approaches, such as using (potentially simple) expression outlier-detection methods, to classify over/under activity? The tertile approach seems poorly suited to the large number of genes that show relatively little variation across samples, or that are either not expressed or expressed in a small fraction of samples. It seems that these thousands of genes would also be examined for potential "co-aberrations", further inflating the large multiple-testing challenge posed by the sheer number of comparisons performed by the pipeline. In any event, it would seem incumbent on the authors to provide a sensitivity/robustness analysis to show how their results are affected if they take top 25%/20% etc. of genes instead of top tertile.

Thanks. We have conducted the suggested robustness analysis. Specifically, we rerun INCISOR by using a 25% threshold for gene activation (i.e, top 25% of samples were considered over-expressed, while the bottom 25% were considered down-regulated and the remaining 50% were considered as WT). The results are shown in figure R4 for the SR-DU case (the results for SR-DD show a similar trend). The overlap for two thresholds is highly significant (Hypergeometric $P < 2.2E-16$). We have added this result in main text (page 6) and described the details in Appendix section 2.4.

Figure R4: Robustness of the SR inferred to varying the activation threshold: Overlap of the SRs predicted (DU) by INCISOR using either gene activation thresholds of 0.33 and 0.25.

6) The authors use of SCNA and gene expression data for different steps in the INCISOR pipeline seems inconsistent-or at least it is confusing. On p. 23, they state that "We divide cell lines into those having high or low expression of gene R... and then "pairs that past both (functional) tests either using gene expression OR SCNA are referred to as putative SR. So here, they take the UNION of gene expression and SCNA. But then in using the TCGA data on p. 24, they mandate that a gene be both over-expressed and have an SCNA (i.e., the intersection of these two sets). Why do they use both one place and either in the other?

Thanks for calling our attention to this issue. Indeed we applied "OR" in step 1 and applied "AND" in step 2 and 3 of INCISOR. "OR" condition was used in step 1 because applying the stringent "AND" condition in step 1, that is, requiring that the putative SR pair must be significant in both SCNA and gene expression in *in vitro* datasets, results in the removal of many SR pairs that have been actually reported in the literature. This is likely to be an artifact arising since many *in vitro* datasets are simply missing SCNA information for many of the genes. In difference, the "AND" condition could be applied consistently in steps 2 and step3 because the TCGA collection does have SCNA information

on all genes for all the patient samples analyzed. This is now explicated in the main text in the methods section, on page 23, as above.

7) The different SR networks differ significantly in sparsity-at least by eye. Can the authors explain why this might be?

Indeed, as the referee pointed out, the SR-DD network is denser than the SR-DU one. As the derivation of these networks uses a shared pipeline in a similar manner, one may assume cautiously that this may reflect an underlying increased abundance of DD vs DR rescue interactions in cancer, but we think it's too early to call on that. To better visualize these findings, however, we decided to add a supp. Figure R5, which shows the degree distributions of current DU and DD networks. In the figure, we refer to the number of edges connected to a vulnerable gene as its in-degree, and number of edges connected to a rescuer gene as its out-degree, viewing an event where gene R rescues gene V as a directed R->V edge. This figure is now added to the Appendix section 3.6 as Figure S2a-d.

Figure R5. Degree distribution of DU (left) and DD (right) networks. Number of edges of vulnerable genes (top row) is referred to as their in-degree and the number of edges of rescuer genes (bottom

row) is referred as their out-degree (viewing the an event where gene R rescues gene V as a directed R->V edge).

8) The authors state that they used 2200 genes in their shRNA in the H/N cancer line HN12. What was the rationale for the choice of 2200 as opposed to entire transcriptome? What are the properties of the genes in this library?

We used a kinase and phosphatase targeted library, as kinases are the most frequent intracellular drug targets and hence focusing on them made sense from a translational perspective. This was originally conducted in collaboration with Ji Luo, at the NCI, who helped develop the shRNA screen pipeline (7). Following the referee's comment, a brief explanation of this motivation has been added to the main text (page 8), stating that "Because kinases are the most frequent intracellular drug targets, we used a kinase and phosphatase targeted library for performing knockdowns bearing their translational relevance."

Also, they state that they infected at an MOI of 1. Why? This is too high-at MOI of 1, a significant number of cells should receive multiple hairpins. Typically, such screens are done at MOI=0.3 or lower.

We agree and thanks for this comment. We indeed used an MOI of approximately .3; the MOI of 1 was in reference to the initial titration in HEK293 cells, which has higher infection efficiency. We have corrected this in the text and thank the reviewer for pointing this out. The revised text (page 28) now reads: "HN12 cells were infected with a library of retroviral barcoded shRNAs at a representation of ~1,000 and a multiplicity of infection (MOI) of ~0.3, including at least two independent shRNAs for each gene of interest and controls."

In addition, inadequate information is given for us to evaluate the validity of this screen-especially the in vivo component. It is well known that only a fraction of the cells that are xenografted actually engraft, which creates a natural "bottleneck" on the shRNA pool. If they just inject cells infected with a neutral barcode library, how many barcodes do they recover (i.e., without any selection)? Put another way, what is the tumor-initiating frequency of this line? Without such information, it is impossible for us to assess this in vivo screen.

Thanks and agreed. In preliminary studies we observed that the tumor initiating efficiency of these cells is very high such that most if not all of the cells will grow after a short term latency. However, detailed analysis and confirmatory experiments with neutral barcoded libraries, for example, will be

required to validate this approach, as appropriately suggested. As such, we agree that the in vivo results can be safely removed, considering that we have not used the information generated by this approach for our computational studies or its validation. We appreciate the reviewer's insightful comment and removed these results from the text, accordingly.

9) Also related to the screen, in Figure 2a, the authors show the precision and recall at the threshold where INCISOR identifies 75 genes-why did they pick this number? As written, the reason for this choice is obscure (at least to me).

The number 75 was arbitrarily chosen simply for illustration purposes. We wanted to show that even at high recall, INCISOR exhibits a reasonable precision. This is illustrated in this example showing that even at high recall (around 70%), where INCISOR predicts 75 genes as SR, it still exhibits a precision of 30%. We have clarified this in the main text (page 8) as:

“INCISOR exhibits a reasonable precision also at high recall rates, e.g., at a threshold when INCISOR predicts 75 genes (recall of about 70%), it still obtains a precision level of 30% (vs 2.1% that is expected by random).”

10) In Figure S2g and h, the authors purport to show the functionally active SR genes "as cancers progress" but it is not clear how they are defining progression-and what are the units on the X axis here? In Figure S2i, they argue that tumors with a high number of SRs, which they define as top 10%, do better than those with lowest 10%--again, some sort of robustness analysis is needed-what happens if they pick highest/lowest 5%, 15%, 20%?.

To study the functional activation of SRs as cancer progresses we divided the breast cancer patients in the METABRIC dataset into 6 classes of cancer progression (removing censored data), by dividing them equally into 6 bins according to their survival times (N=627). First, in each bin, we counted the mean fraction of functionally active SRs, by integrating the expression and SCNA data of the samples it includes to this end. Second, we defined a vulnerable gene as rescued if more than a threshold number of rescuers are over-activated and subsequently, we count the mean fraction of rescued vulnerable genes in the six progression bins. Following your comments, we have now re-run the survival analysis to examine the results considering a few options for defining the sets of top-rescued vs bottom-rescued samples. We conducted a Kaplan-Myer analysis (Fig. R6) with different such cut-offs to check their robustness, as shown in Fig S2i. Specifically, in addition to threshold 10% of the samples to define the extreme sets, we tested thresholds of 5%,15%, and 20%. As shown in figure R6(a) the respective results are: (i) 5% ($\Delta\text{AUC}=0.27$ $P=2.6\text{E}-06$), (ii) 15% ($\Delta\text{AUC}=0.12$ $P=1.5\text{E}-06$) and (iii) 20% ($\Delta\text{AUC}=0.17$ $P=1.5\text{E}-10$), testifying to the overall robustness of the trends reported.

We now refer to these results briefly in the main text (on page 6) and describe them in more detail in Appendix section 3.2.

.. Also, in Figure S2j, they compare survival combining SR and SL, but they don't show us survival with just SL-why not?

Following on the reviewer's request, we have now added a figure (R6(b)) to describe the survival prediction curves obtained with a parallel, SL analysis. This have been added Appendix section 3.3.

Figure R6: **(a) Robustness of the results to variations in the sizes of the top and bottom rescued sets analyzed in the KM analyses:** The KM analysis compares the survival of patients whose tumors have many functionally active SRs (*rescued*) to those with a few (*non-rescued*). Different thresholds (5%, 15% and 25% of the samples) were used to define the extremum rescued vs non-rescued sets of samples. The differences in the survival (area under the curve (Δ AUC) and their log rank p-values) between rescued and non-rescued samples are denoted at each threshold, with arrows connecting between survival curves of matching top and bottom sets obtained at a given threshold. **(b): Survival prediction of SL (ISLE (8)):** The KM plots shows that patients with high SL-scores (solid lines) have

better prognosis than those with low SL-scores (dashed lines), using five different SL score thresholds, namely top/bottom 10% (red), 20% (orange), 30% (green), 40% (blue), and 50% (purple).

11) It is not clear what the authors are trying to convey with Figure S2L-this figure shows substantial differences in the number of SRs in different cancers-but what is their argument-surely SRs are not the only determinant of response, as for many of these cancers, whether or not they are drug sensitive is dispositive. For example, ovarian cancer might have one behavior unperturbed-which could depend on SRs for the genetic changes that cause the tumor-but whether or not a given ovarian cancer patient survives or dies is largely dependent on whether they have HR deficiency (and thus are cis-PLT sensitive) or not. Similarly, whether a breast cancer patient survives or not can depend on HER2 and ER status.

Obviously we agree that SRs are just one of the factors that contribute to response – naturally, we focus here on studying SR's role and its potential importance (while controlling for numerous potential confounding factors). Figure S2L is aimed at demonstrating that the predicted SR interactions are associated with survival across multiple cancer types, i.e. they are likely to be clinically relevant across multiple cancer types. This is further illustrated in Dataset Table S2 and S4, where we evaluate the clinical significance of each individual SR interaction in each cancer type. As evident most of (both DU and DD) predicted SR interactions are found to be clinically significant in more than one cancer type.

12) On p. 8, the authors test a number of drug combinations predicted by their SR networks, and claim that several are synergistic. However, two of the combinations involve dasatanib, which inhibits MULTIPLE kinases, not just KIT and SRC as they imply. How can such a broad spectrum kinase inhibitor be used in any way to infer specificity. Similarly, their PTK2 inhibitor also inhibits PTK1-how can they infer specificity. Similar problems attend the use of other very broad spectrum kinase inhibitors, such as Sunitinib and Sorafenib, which are most certainly NOT (only) VEGFR and RAF inhibitors, respectively.

Thanks. We agree that we cannot discount the possibility that the observed synergism could be a result of multiple drug targets. This potential caveat has been addressed in the experiments we conducted in the NSCLC cell-lines (Fig 3a), where we extensively tested a large compendium of drugs and cell lines to study the synergism between drug pairs targeting primary targets and their predicted SRs. In fact, we chose to experimentally test the SR interactions involving DNMT1 because of availability of multiple known drugs that could target predicted SR partners of DNMT1 with high

specificity. As reported in the main text on page 9, these results confirm the predicted synergism between a DNMT1 inhibitor and multiple inhibitors of its SR partners across multiple cell lines. Furthermore, drugs targeting DD-SR genes were antagonists in majority of the 18 cell lines tested, as predicted. We have now clarified this in the main text on page 9 and in the discussion page 17 as: “Because of many of these drugs tested above are known to target multiple genes, we conducted additional experiments in NSCLC to confirm the relationship between synergism and predicted SRs.” “...To discount the possibility that observed synergism is not due to non-specific targeting of drugs, including rapamycin, we conducted siRNA and a large-scale drug combination screen.”

We think that the experiments involving Dasatanib are still of value but if the referee still think otherwise we can move them to Supp. Material or remove them all together.

13) Related to point 10, the authors state that rapamycin is a TOR inhibitor, but it is only a TORC1 inhibitor-Torin would be a better choice to test if one is claiming to represent TOR inhibition.

Our pertaining experiments have been motivated by reports that long term treatment with rapamycin has been shown to inhibit mTORC2 in various cellular systems (9). Indeed, we (the Gutkind lab) have shown this to be the case in HNSCC, in which we see evidence of mTORC2 inhibition after 2 days of treatment with rapamycin in numerous HNSCC experimental models(10-12), and after short term treatment of HNSCC patients with rapamycin (13). We have clarified this issue in the methods (page 28). Nonetheless, we have also clarified that rapamycin blocks preferentially mTOR in its mTORC1 complex, and reinforce this concept in the discussion section (page 17), which now reads:

“To validate the rescue effect due to mTOR inhibition, we used Rapamycin that blocks preferentially mTOR in its mTORC1 complex.”

14) The authors state that "to predict the response of an individual patient's (sic) to a given drug, we defined the drug-tumor SR score as the number of upregulated rescuers of a drug's target in that patient's tumor.." OK-but why the number? Wouldn't the strength of the SR be relevant-i.e., are two strong SRs better than 100 weak ones? Why should it just be the number-or better still, why don't the authors provide some type of analysis justifying this choice?

Thanks for this insightful suggestion. Subsequently, we have now checked if using the strength of SR interactions improves drug response prediction. Specifically, as suggested by the reviewer we weight the functionally active rescuers by the SR strength, i.e, the SR score assigned by INCISOR. As shown in Fig R7, this strategy shows significant predictive power for 13 drugs. compared to 14 drugs whose response can be predicted using the unweighted SR counts (where response was estimated in terms of

tumor size reduction). Given these results we decided to keep the main text as is but have now added the report on the weighted-SR results in the Appendix section 5.6 (Appendix Page 29).

Fig R7: **Using SR strength to predict cancer drug response in patients** (in terms of tumor size reduction). The Y-axis denotes the sum of SR strengths (SR-scores) of the DU-SR upregulated rescuers in tumors of responders (orange) and non-responders (green) for each drug (X-axis). Significant results are marked by stars.

15) In Figures S4f, g, what does "normalized IC50" mean-how is this normalized?

For a drug, we first convert its IC50 values across cell lines to their corresponding quantiles (*qIC50*). “normalized IC50” is then defined as the mean of the *qIC50* in conditional case (defined as cell line with an inactive vulnerable gene). For example, normalized IC50=80% implies that conditional IC50 is larger than IC50 of 80% of all cell lines. This was done because the baseline IC50 of different drugs differs a lot, so the *qIC50* allows to compare the effects of vulnerable gene inactivation on different drugs. We have clarified this issue now in Appendix section 4.3 (page 21) - thanks.

Also, I don't understand the data presentation in Figure S4h. The legend says, "The phenotypes measured are the post-knockdown growth rates in pooled shRNA screening." What do they mean? What one measures in pooled screens is enrichment for, or depletion of, barcodes representing different shRNAs-one doesn't measure growth rate. Did they test some number of hits from the screen and measure growth rate? Please explain.

Thanks for pointing out the wrong terminology in Figure S4h, for which we sincerely apologize. The figure simply compares the essentiality of rescuer genes in cell lines where the vulnerable gene is inactivated to those where it is not. We have accordingly replaced the "cell growth" with "essentiality" in the Figure S4h.

16) The validations in the TOR experiments (and others) are sub-optimal. We should at least see a few cases where they did knockdown and rescue (the only compelling proof) or an orthogonal type of experiment like CRISPR/Cas9 deletion or dominant negative. At the very least, they should show knockdown efficiency so we can see that the biological effect parallels the extent knockdown.

We have extensively validated the knock down efficiency of the kinases shRNA library in our prior studies (7), but following your comment we now have further validated the knock down efficiency of PI3K and mTOR, which were used in the combinatorial drug evaluation analysis. These results are now reported in Figure R8. We now briefly refer to them briefly in the methods section (page 28 and 30). The main text reads:

“..Knock down efficiency of the kinase shRNA library was validated in our prior studies..”

“ .. We also validated the knock down efficiency of PI3K and mTOR siRNA via a western blots analysis (Fig S11h) as follows. Cells were transfected with negative control or the corresponding PIK3CA, mTOR (FRAP1) siRNAs. Cells were lysed in lysis buffer (50 mM Tris-HCl pH 7.6, 150 mM NaCl, 1 mM EDTA, 1% Nonidet P-40) supplemented with Halt™ Protease Inhibitor Cocktail (Thermo Fisher scientific) and phosphatase inhibitors (1 mM Na3VO4 and 1 mM NaF). Equal amounts of total proteins were subjected to SDS-polyacrylamide gel electrophoresis. Primary antibodies used were from Cell Signaling Technology (Danvers, MA), PI3KCA (catalog number 4255), pAKT (catalog number 2965), AKT (catalog number 9272), pmTOR (catalog number 5536),

mTOR (catalog number 2983), pS6 (catalog number 2211), S6 (catalog number 2217), and GAPDH (catalog number 2118), the latter as a protein loading control.“

Figure R8. Western blot analysis of signaling events in HNSCC after knock down of the *PIK3CA* and mTOR. (A) Top, HN12 (left) and SCC47 (right) cells were transfected with negative control (C) or the corresponding *PIK3CA* siRNAs for 72 hours, and lysates were analyzed as indicated (Methods). (B) Bottom, HN12 (left) and SCC47 (right) cells were transfected with negative control (C) or the corresponding mTOR (*FRAP1*) siRNAs for 72 hours, and lysates were analyzed as indicated. In every case, the ‘-’ indicates control cells without transfected siRNA.

17) What are the authors trying to convey in Figure S10h/I when none of the differences appear to be statistically significant?

We wanted to illustrate that, although not statistically significant, the expression of some of the rescuers is altered in the predicted direction already in on-treatment tumor biopsies. This suggests such patients might benefit by targeting the predicted rescuer. Because this hypothesis requires more rigorous testing, we have only included these result in the Appendix figure as suggestive for potential future follow ups. We still think it could be worthy and of interest to our readers, but we can omit it in the final version if preferred by the reviewer.

Minor comments:

1) There are inconsistencies in nomenclature at various places in the paper: e.g., sometimes the authors use DD-SR, other times DD-SR. Please be consistent.

Thanks. Changed everything to DD-SR consistently.

2) Similarly, in some of the figure headings, all words are capitalized; in others, just the first one is. We have changed the figure captions.

3) The authors start several sentences with numbers-please type our the number if it begins a sentence.

Thanks, changed accordingly in main text and Appendix.

4) On p. 4, the authors describe DU-SR genes first and then DD-SRs, yet in the figure, they show DD in middle and DU on right-I would reverse this order

Thanks. Changed the order as suggested.

5) It is difficult to read the X and Y axes on many of the figures; it is also difficult to read some of the figures (especially the interaction diagrams). Please print out every figure and make certain that each can be read easily by someone with 20:20 vision (not X-ray vision!).

We have changed figures and font size for clarity.

6) Also, if one clicks on a node in a network diagram, one should be able to see the identify of the gene-only some of the genes are labelled in their network diagrams, and the rest are seen only as spots. Please correct this.

With more than 1000 nodes it is difficult to include this feature in figures. In response to your comment, we have now included an interactive figure that can be opened using Cytoscape with this feature.

7) On p. 8, they authors state that mTOR can activate AKT independently of PI3K, and cite a review as a reference-as this is certainly NOT generally believed to be the case, please cite the primary reference.

AKT is a typical target of the mTORC2 complex (9). Specifically, mTORC2 regulates phosphorylation of AKT. We have added the reference to the text.

8) On p. 10, the authors state that DNMT1 is a "key oncogene in non-small cell lung cancers." DNMT1 is over-expressed in many such tumors, but I am unaware of any evidence showing that it meets the criteria for an oncogene-please provide a reference to support this claim. Also, the pathways for DNMT1/b-catenin, etc., are not all well validated and should probably be presented as plausible or possible pathways to explain the data, not gospel.

We simply meant DNMT1 as a "key cancer gene". We apologize for not adhering with precise terminology and creating the confusion. DNMT1 was chosen for further analysis in the experiment because INCISOR predicted it as a major hub that has SR-interactions with numerous genes that could be targeted by drugs. We have modified the text accordingly to read (page 9): "We picked DNMT1 to test this hypothesis as it is a major hub in the DU-SR network (Figure 1c) and a key cancer gene that is predicted to have SR interactions with multiple genes that can be targeted by drugs."

9) Figure 3 D/E - Given the discrete nature of these multinomial distributions, each consisting of a small number of possible integer values (1-6), a boxplot representation useful for continuous (or many-valued discrete) distributions seems inappropriate. A more suitable visualization may be side-by-side bar charts. Also, "protein" is mis-spelled in Figure 3D.

Thank you. Following up on your suggestion we have updated the figure (Fig R9) and added it to the main text (Fig 3):

Fig R9: Functional similarities between gene pairs in the DU-SR network. Comparison of functional similarities between interactions in (i) the DU-SR network (ii) Random-pairs: the network is generated by random pairing between protein-coding genes, having a degree distribution similar to that of the DU-SR network (iii) shuffled-pairs: the network is generated by shuffling pairing of the DU-SR network. Functional similarities of genes in each pair were evaluated in terms of: (a) the distance between the paired genes in human PPI network (14) and (b) these distances in the STRING network (15) (where the distance denotes the number of interactions on the shortest path between the paired genes). The histogram of network distances between gene pairs are displayed for the PPI and STRING networks.

10) The legend to Figure S2n makes no sense as written. Please correct

Thanks. Done.

11) For Figure S3, please label the drugs on the figure-what is "BRAF inhibitor?"

Thanks. Done.

Also, what is "estrogen receptor inhibitor."

In Zhang et. al. (6) multiple shRNA targeting estrogen receptor were referred as “estrogen receptor inhibitor”.

Do they see differences between Vemurafenib and "BRAF inhibitor" in V600E melanoma lines-and if so, can they explain why? Ditto for tamoxifen and estrogen receptor inhibitor in ER+ breast cancer (i.e., can they find SRs that are different for these two drugs)?

INCISOR cannot identify different SRs for drugs targeting same target genes. Also, the current version INCISOR does not predict SRs that are specific to a specific cell line or condition).

To identify SR specific to V600E cell lines would require applying INCISOR to subset of cell lines and patient samples with V600E mutations of sufficient size, which is beyond the scope of the current work. We now relate to this point in the discussion section of the manuscript on page 17, which reads “Finally, as this is the first genome-wide study of cancer SR interactions, we focused on identifying SRs that are common across many cancer types. INCISOR identifies the same interactions for all drugs targeting the same gene(s). Future studies, however, will aim to identify cancer type-specific and context-specific SR networks as more data accumulates”.

12) The authors forgot to plot the "Random" line on Figure S4B.

Thanks. Added the line.

References

1. C. Curtis *et al.*, The genomic and transcriptomic architecture of 2,000 breast tumours reveals novel subgroups. *Nature*, (2012).
2. C. Y. IFong *et al.*, BET inhibitor resistance emerges from leukaemia stem cells. *Nature* **525**, 538-542 (2015).
3. P. Rathert *et al.*, Transcriptional plasticity promotes primary and acquired resistance to BET inhibition. *Nature* **525**, 543-547 (2015).
4. T. J. Stuhlmiller *et al.*, Inhibition of Lapatinib-Induced Kinome Reprogramming in ERBB2-Positive Breast Cancer by Targeting BET Family Bromodomains. *Cell Rep* **11**, 390-404 (2015).
5. J. R. Mills *et al.*, RNAi screening uncovers Dhx9 as a modifier of ABT-737 resistance in an Eμ-myc/Bcl-2 mouse model. *Blood* **121**, 3402-3412 (2013).
6. Y. W. Zhang *et al.*, Acquisition of estrogen independence induces TOB1-related mechanisms supporting breast cancer cell proliferation. *Oncogene* **35**, 1643-1656 (2016).
7. L. C. Lee, S. Gao, Q. Li, J. Luo, Using pooled miR30-shRNA library for cancer lethal and synthetic lethal screens. *Methods Mol Biol* **1176**, 45-58 (2014).

8. J. S. Lee *et al.*, Harnessing synthetic lethality to predict the response to cancer treatment. *Nat Commun* **9**, 2546 (2018).
9. M. Laplante, D. M. Sabatini, mTOR signaling in growth control and disease. *Cell* **149**, 274-293 (2012).
10. P. Amornphimoltham, V. Patel, K. Leelahavanichkul, R. T. Abraham, J. S. Gutkind, A retroinhibition approach reveals a tumor cell-autonomous response to rapamycin in head and neck cancer. *Cancer Res* **68**, 1144-1153 (2008).
11. D. Martin *et al.*, The head and neck cancer cell oncogenome: a platform for the development of precision molecular therapies. *Oncotarget* **5**, 8906-8923 (2014).
12. R. Iglesias-Bartolome *et al.*, mTOR inhibition prevents epithelial stem cell senescence and protects from radiation-induced mucositis. *Cell Stem Cell* **11**, 401-414 (2012).
13. Z. Wang *et al.* (AACR, 2017).
14. M. H. Schaefer *et al.*, HIPPIE: Integrating protein interaction networks with experiment based quality scores. *PLoS One* **7**, e31826 (2012).
15. D. Szklarczyk *et al.*, STRING v10: protein-protein interaction networks, integrated over the tree of life. *Nucleic Acids Res* **43**, D447-452 (2015).

Thank you again for submitting your revised work to Molecular Systems Biology. We have now heard back from the two referees who accepted to evaluate the revised study. As you will see they are globally supportive. They raised however some remaining points which we would kindly ask you to address in a revision of this work:

- reviewer #2 feels that the siRNA validation of the PI3KCA x mTOR pair should be better controlled with, for example, CRISPR/CAS9 deletions to rule out off-target effects. We agree that this would be ideal. On the other hand, we also recognize that the use of drug inhibitor pairs and drug-siRNA pair represents, to some extent, (semi-)orthogonal assays, where off-target effects of siRNA are unlikely to be identical as off-target effects of the corresponding drugs. However, in view of the lack of specificity of dasatinib, we would ask you to either provide more solid evidence from CRISPR experiments or to tone down the conclusions of these experiment and only state that the results of these experiments are "consistent" with the interpretation given.

- Some additional clarifications are requested by reviewer #2 and the recommendations are very clear in this regard.

REFEREE REPORTS

Reviewer #1:

I thought that the authors had done a good job revising the manuscript and my concerns were addressed. I support acceptance for publication.

The manuscript has grown in complexity due to this revision and a large amount of material and supplementary material is now included. The authors may have a careful look at improving clarity.

There are minor copy-editing issues in the manuscript, for example, the reference "Cancer Genome Atlas Research Network. Electronic address & Cancer Genome Atlas Research, 2017;" right on Page 2.

Reviewer #2:

This is a revised version of a manuscript that I reviewed earlier, which develops and validates a new pipeline, INCISOR, to detect "synthetic rescue" interactions. The other reviewer and I were generally enthusiastic about the concept of the paper, but had a number of concerns, most of which centered around issues of robustness of the pipeline, controls for multiple comparisons, the choice of cut-offs for judging "high and low," and specific issues regarding the specificity of some of the methods used to test proposed SR interactions.

In the revision, the authors have made a good faith effort to address these concerns, and the manuscript is improved substantially. There are, however, a few remaining issues.

1) I think that the authors missed my point about the validation of the sh/siRNAs. The point is not whether these reagents deplete the target, it's whether the observed effect is a consequence of that depletion and not an off-target effect. To this end, at least some predictions **MUST** be validated with either an orthogonal method (CRISPR/CAS9 deletion, dominant negative) or by rescue with an sh/si-resistant expression construct. Even the use of two independent siRNAs is really not adequate, as there are definitely cases where these can have off-target effects.

2) The authors, in their response to reviewers letter, seem to equate "cancer progression," with time of survival (i.e., they assume a cancer is more "progressed" if the patient lives for less time). "Progression" (a concept that I deplore, anyway), usually means acquisition of more cancer-causing mutations and/or acquisition of more cancer-associated properties (e.g., metastasis or immune evasion). The reasons for patient survival are much more complex-and do not necessarily affect "progression," defined as above. Instead, it could just be the lack of a well-defined, effective therapy. Instead of using loose terms, they should just be explicit about what they mean.

3) I'm sorry, but dasatinib inhibits so many kinases, that I cannot see how this agent can be used to validate ANY specific SR interaction. I think that these data should be removed.

4) Similarly, if none of the differences in Figure S10h/I are statistically significant, they shouldn't be

included as figures. If the authors wish, they can state that several trended towards, but did not reach, statistical significance.

5) The revision has multiple typos, font changes, and mis-spellings. More careful editing of the final version is advised.

Editor:

- reviewer #2 feels that the siRNA validation of the PI3KCA x mTOR pair should be better controlled with, for example, CRISPR/CAS9 deletions to rule out off-target effects. We agree that this would be ideal. On the other hand, we also recognize that the use of drug inhibitor pairs and drug-siRNA pair represents, to some extent, (semi-)orthogonal assays, where off-target effects of siRNA are unlikely to be identical as off-target effects of the corresponding drugs. However, in view of the lack of specificity of dasatinib, we would ask you to either provide more solid evidence from CRISPR experiments or to tone down the conclusions of these experiment and only state that the results of these experiments are "consistent" with the interpretation given.

Thanks. We have toned down claims regarding Dasatinib and siRNA experiment accordingly, stating that they are consistent with our interpretation (page 9, 2nd paragraph), and moved the discussion regarding Dasatinib (removed from Fig 2) to the Appendix as follows:

“In the third experiment, we conducted siRNA experiments to show that observations of Figure 2b are consistent. Targeting mTOR by siRNA exhibited enhanced sensitivity to BYL719 in 4 of these cell lines (Appendix 4.7, Fig. S8). Similarly, siRNA targeting of PIK3CA exhibited enhanced Dasatinib sensitivity (Appendix 4.7, Fig. S8).”

- Some additional clarifications are requested by reviewer #2 and the recommendations are very clear in this regard.

We have modified the revised version to address all of the remaining recommendations.

Reviewer #1:

I thought that the authors had done a good job revising the manuscript and my concerns were addressed. I support acceptance for publication.

Thanks!

The manuscript has grown in complexity due to this revision and a large amount of material and supplementary material is now included. The authors may have a careful look at improving clarity.

We have carefully revised the manuscript to improve its clarity.

There are minor copy-editing issues in the manuscript, for example, the reference "Cancer Genome Atlas Research Network. Electronic address & Cancer Genome Atlas Research, 2017;" right on Page 2.

Thanks for pointing this out. We have corrected the citation.

Reviewer #2:

This is a revised version of a manuscript that I reviewed earlier, which develops and validates a new pipeline, INCISOR, to detect "synthetic rescue" interactions. The other reviewer and I were generally enthusiastic about the concept of the paper, but had a number of concerns, most of which centered around issues of robustness of the pipeline, controls for multiple comparisons, the choice of cut-offs for judging "high and low," and specific issues regarding the specificity of some of the methods used to test proposed SR interactions.

In the revision, the authors have made a good faith effort to address these concerns, and the manuscript is improved substantially. There are, however, a few remaining issues.

Thanks!

1) I think that the authors missed my point about the validation of the sh/siRNAs. The point is not whether these reagents deplete the target, it's whether the observed effect is a consequence of that depletion and not an off-target effect. To this end, at least some predictions **MUST** be validated with either an orthogonal method (CRISPR/CAS9 deletion, dominant negative) or by rescue with an sh/si-resistant expression construct. Even the use of two independent siRNAs is really not adequate, as there are definitely cases where these can have off-target effects.

Following up on the editor's suggestion, rather than prioritize further CRISPR experimental validations we have toned down claims regarding siRNA experiments (and the experiments concerning Dasatinib as well following your third comment).

2) The authors, in their response to reviewers letter, seem to equate "cancer progression," with time of survival (i.e., they assume a cancer is more "progressed" if the patient lives for less time). "Progression" (a concept that I deplore, anyway), usually means acquisition of more cancer-causing mutations and/or acquisition of more cancer-associated properties (e.g., metastasis or immune evasion). The reasons for patient survival are much more complex-and do not necessarily affect "progression," defined as above. Instead, it could just be the lack of a well-defined, effective therapy. Instead of using loose terms, they should just be explicit about what they mean.

Thanks for pointing this important distinction out, with which we obviously agree. In the revised manuscript we explicitly made a clear distinction between patient survival and cancer progression (Appendix page 16).

3) I'm sorry, but Dasatinib inhibits so many kinases, that I cannot see how this agent can be used to validate ANY specific SR interaction. I think that these data should be removed.

We have moved the report of the Dasatinib experiments to the supplementary and toned down the conclusions accordingly (page 9, 2nd paragraph) .

4) Similarly, if none of the differences in Figure S10h/I are statistically significant, they shouldn't be included as figures. If the authors wish, they can state that several trended towards, but did not reach, statistical significance.

Thank you, and agreed; accordingly, we have removed the supplementary figure S10h/i.

5) The revision has multiple typos, font changes, and mis-spellings. More careful editing of the final version is advised.

We have carefully reviewed the revised version and edited any typos, grammar and formatting issues that we could find.

Accepted

18th January 2019

Thank you again for sending us your revised manuscript. We are now satisfied with the modifications made and I am pleased to inform you that your paper has been accepted for publication.

Corresponding Author Name: Eytan Ruppin, Aviansh Das

Journal Submitted to: Molecular System Biology

Manuscript Number: MSB-18-8323RRR